# UTILITY-DIRECTED CONFORMAL PREDICTION: A DECISION-AWARE FRAMEWORK FOR ACTIONABLE UNCERTAINTY QUANTIFICATION

**Santiago Cortes-Gomez, Carlos Patiño**[*]**, Yewon Byun, Zhiwei Steven Wu, Eric Horvitz**[†] **& Bryan Wilder**
Machine Learning Department, Carnegie Mellon University
[*]University of Amsterdam
[†]Microsoft
{scortesg, yewonb, zstevenwu, bwilder}@cs.cmu.edu
carlos.patino.paz@student.uva.nl
horvitz@microsoft.com

## ABSTRACT

Interest has been growing in *decision-focused* machine learning methods which train models to account for how their predictions are used in downstream optimization problems. Doing so can often improve performance on subsequent decision problems. However, current methods for uncertainty quantification do not incorporate any information about downstream decisions. We develop a methodology based on *conformal prediction* to identify prediction sets that account for a downstream cost function, making them more appropriate to inform high-stakes decision-making. Our approach harnesses the strengths of conformal methods—modularity, model-agnosticism, and statistical coverage guarantees—while incorporating downstream decisions and user-specified utility functions. We prove that our methods retain standard coverage guarantees. Empirical evaluation across a range of datasets and utility metrics demonstrates that our methods achieve significantly lower costs than standard conformal methods. We present a real-world use case in healthcare diagnosis, where our method effectively incorporates the hierarchical structure of dermatological diseases. The method successfully generates sets with coherent diagnostic meaning, potentially aiding triage for dermatology diagnosis and illustrating how our method can ground high-stakes decision-making employing domain knowledge.

## 1 INTRODUCTION

Uncertainty quantification in classification problems is increasingly addressed through a model-agnostic technique known as *conformal prediction* Vovk et al. (2005); Angelopoulos & Bates (2021). Instead of outputting a single label and confidence, conformal prediction outputs a set of outcomes with guaranteed statistical coverage, ensuring that the true outcome is included in the set with a pre-specified, target confidence level. This procedure for quantifying uncertainty through inference about sets of outcomes, has been leveraged to enable safer decision-making based on machine learning model outputs. As an example, in dermatology diagnosis, a conformal analysis might predict that the patient's illness is a member of a set of possible mutually exclusive classes, such as [eczema or psoriasis]. That is, a guarantee is provided that the true condition is highly likely to be included in the predicted set, making inference about the set a more robust estimate than raw model outputs about singular outcomes. Recent efforts have pursued uses of conformal prediction to develop AI systems that can support decisions in high-stakes situations (Vemula et al., 2018; Dvijotham et al., 2023; Yin et al., 2022), where the decisions restricted to actions within the prediction set are deemed as safe.

We note that statistical coverage of the true label is not always a strong enough desiderata to guarantee *informative* sets for decision making. In many applications, prediction sets should ideally be aligned with a utility function that captures preferences about downstream decisions and outcomes. For instance, in medical diagnosis, a predicted set of diseases may be more actionable if the different suggested diagnoses share the same treatment, costs and benefits of treatments, or if it is straightforward to conduct additional tests for each diagnosis in the set to discriminate among them.

Indeed, there has been a growing body of work that bridges machine learning predictions and decision problems modeled by optimization programs. Among this literature there is growing interest for the so called *decision-focused* machine learning (Wilder et al., 2019; Mandi et al., 2020; Wang et al., 2021; Shah et al., 2022; Elmachtoub & Grigas, 2022; Ren et al., 2024), where models are trained by maximizing a downstream optimization problem in the core loop of machine learning. In contrast, conformal methods are designed to be agnostic to the model, loss function, and downstream decision task; their generality is a key feature that allows this method to be used flexibly as a post-processing step that is completely decoupled from the model.

Inspired by the spirit of decision-focused learning, we introduce methods that bridge the strengths of conformal approaches with the goal of incorporating information about downstream decision problems, retaining the modularity and ease of use of conformal prediction, while enabling users to align prediction sets with task-specific utility functions. We are not proposing an end-to-end approach to train a model to output predictions that are going to yield higher utility on a downstream application. Rather we show how to generate prediction sets that minimize a user-specified cost function while maintaining a high-probability coverage guarantee. To achieve this, we propose two families of algorithms. The first family, extending existing methods for conformal prediction, incorporates a penalty based on the cost function of a prediction set. This creates a collection of conformal predictors indexed by a hyperparameter which weights cost function (the penalty) against a standard non-conformity score. The hyperparameter is then optimized on a validation set to find the value with minimum cost function. While this method often performs well, it can struggle to adapt to the combinatorial structure of downstream cost function (e.g., non-linear interactions between elements of the prediction set) and adds complexity in the form of a hyperparameter to tune. Therefore, to address these limitations, we introduce a second family of algorithms which instead solve an optimization formulation whose solution is the set with the minimum loss subject to statistical coverage constraints, i.e, the sought after predicted set. Algorithmically, this is instantiated by solving a lightweight optimization problem for each test instance, followed by post-processing a function of the base classifier's predictions to enforce the desired statistical coverage. We prove that both approaches provide standard coverage guarantees and find empirically that they are often able to find sets with significantly smaller costs than standard conformal methods.

Incorporating the cost function into uncertainty quantification allows us to address the goal of producing more *informative* prediction sets. For instance, in the aforementioned case study on dermatology diagnosis, we show how specifying a natural cost function (a coverage function) enables us to generate prediction sets with more clinically coherent interpretation. Figure 1 provides an illustrative example of a prediction set output by a standard conformal approach compared to that of our method. We observe that the different labels within the standard conformal prediction set would imply completely different downstream actions for diagnosis and treatment, as benign vs. malignant diseases need to be dealt with differently. Even though the set contains the true label, it is not informative for decision making. By contrast, the prediction set output by our method contains diagnoses within a common family of malignant epidermal diseases, while satisfying the same coverage guarantees. This hierarchical homogeneity of the prediction set aids clinicians in making follow-up decisions (e.g., reflecting about the disease process category at hand, prioritizing follow-up tests, prescribing treatments, etc.). We accomplish this by specifying a cost function (detailed in Section 3.2) based on coherence of the prediction set with respect to an expert-defined hierarchy for dermatological diagnosis. Such hierarchical abstractions are a common feature of medical diagnosis and many other real-world classification tasks, and expert decision making more broadly, and a valuable characteristic of our method is that it can be employed to align prediction sets with the abstractions that human experts use to make decisions based on predictions.

To summarize, our contributions are as follows:

- We develop three methods for constructing conformal predictors which generate sets with low average cost function based on a pre-defined utility function, thereby producing more desirable sets from the perspective of downstream use. We start by considering cost functions which are *separable*, meaning that they can be decomposed into the sum of a penalty for each label included in the prediction set. We then generalize these ideas to the non-separable case, where the cost function may contain nonlinear interactions between elements of the set. In each case, we prove that our algorithms produce sets with the desired level of statistical coverage.

- We conduct experimental analyses across several datasets, including both standard image classification datasets and a real-world case study in dermatological medical diagnosis. Our proposed algorithms result in prediction sets with improved costs across all settings, often by a substantial amount,

e.g., reductions of 60-75% in loss. Our method successfully captures patterns of homogeneity informed by external domain knowledge. In particular, for the case study, our methods produce prediction sets that have diagnostic meaning, demonstrating how our method leverages uncertainty quantification to ground high-stakes decision-making—in this case in the healthcare domain. Our implementation is publicly available at link.

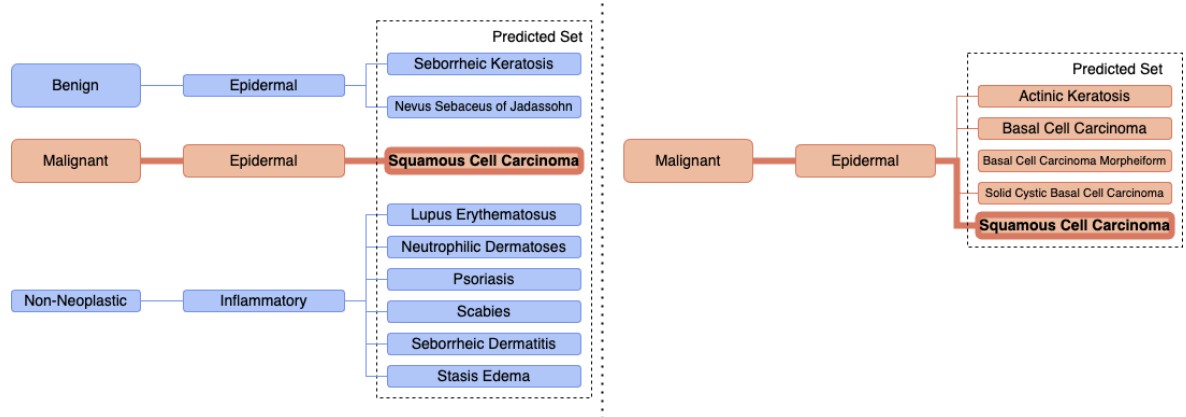

Figure 1: Illustration of how proposed methods compare to the standard formulation of conformal prediction. The real-world example is taken from the Fitzpatrick dataset where the true disease is squamous cell carcinoma. The standard conformal prediction method results in a set of nine elements that span benign, malignant, and non-neoplastic diagnoses. Our method provides a formulation with a set of five elements within the malignant melanoma category.

## 1.1 RELATED WORK

**Decision-Focused Learning** A growing body of literature focuses on aligning machine learning model predictions with downstream optimization objectives, either under the predict-then-optimize framework (Mandi et al., 2020; Elmachtoub & Grigas, 2022; Wang et al., 2021), or using end-to-end approaches like decision-focused learning (Wilder et al., 2019; Shah et al., 2022; Ren et al., 2024). Of particular relevance to our work is Horvitz & Klein (1993), which introduces a utility-theoretic formulation of inference and decision making at progressively higher-levels of abstraction about outcomes (i.e., abstraction is defined in terms of moving from single hypotheses to sets of mutually exclusive outcomes). This work is particularly interesting and motivating to us, as it is proposed as a solution to a similar real-world case study as ours but using ideas related to Bayesian inference framed in the context of a utility-theoretic bounding formulation.

**Uncertainty Quantification and Conformal Prediction** The literature on uncertainty quantification is vast, from Bayesian methods, such as Bayesian neural networks (MacKay, 1992; 1995; Neal, 1996; Wilson & Izmailov, 2020) and ensemble methods (Wasserman, 2000; Lakshminarayanan et al., 2017), to frequentist approaches, such as quantile regression (Koenker & Bassett Jr, 1978; Koenker & Hallock, 2001; Yu et al., 2003; Tagasovska & Lopez-Paz, 2019; Pouplin et al., 2024) and conformal prediction (Vovk et al., 2005; Romano et al., 2020; Angelopoulos et al., 2020b). Quantile regression is particularly noteworthy, as it can be adapted to produce conformal predictors for continuous variables. However, the study of the continuous case lies outside the scope of this work. Of utmost relevance to our work is the literature on conformal prediction.

Conformal prediction has gained popularity in recent years as a model-agnostic, computationally low-cost post-processing method for principled uncertainty quantification Vovk et al. (2005). In particular, Romano et al. (2020) and Angelopoulos et al. (2020b) explore how to create prediction sets whose size adapts to the confidence levels output by the model. When the model is more certain about a given prediction, the prediction set becomes smaller. Specifically, Angelopoulos et al. (2020b) generates adaptive sets that are smaller on average, a task that represents a particular case of the type of problems we aim to solve. Conformal prediction has been used in the context of decision making in the medical setting, for example Lu et al. (2022) propose a a

conformal predictor that provides group conditional guarantees. They are interested in provide prediction sets that are usable across groups in order to ensure fair decisions, thus increasing clinical usability and trustworthiness in medical AI. In other words, this work is focused on guarantee fairness rather than a general formulation of utility-focused prediction sets.

At the intersection of machine learning for decision-making and conformal prediction, and thus most closely related to our work, is conformal risk control Angelopoulos et al. (2022). The approach is a generalization of conformal prediction that allows for controlling the expected value of a desired loss at a preset level. The method can achieve true statistical coverage in multi-label classification or, more generally, for any loss function that decreases as the size of the prediction set increases. Although this method offers flexibility, it does not directly optimize the expected value of the chosen function, and depending on such function, controlling for the expected value of the chosen loss can be at expense of statistical coverage. Notably, our method imposes no restrictions on the type of losses that can be chosen, allowing it to handle cost functions that do not decrease monotonically with respect to set size and for which it is impossible to guarantee an absolute upper bound. We also note there is a wealth of literature that studies uncertainty quantification in similar settings (Qi et al., 2021; Chenreddy et al., 2022; Kallus & Mao, 2023; Sadana et al., 2024; Chenreddy & Delage, 2024; Patel et al., 2024; Yeh et al., 2024).

## 2 PROBLEM FORMULATION

Let $\mathcal{Y}$ a finite set of labels. Define the cost function of a prediction set as a function $\mathcal{L} : 2^{\mathcal{Y}} \to \mathbb{R}$. In our framework, $\mathcal{L}$ evaluates the utility of a given prediction set, where a suitable prediction set for decision-making is one that has a low value of $\mathcal{L}$. Let $P$ be a joint distribution over $\mathcal{X} \times \mathcal{Y}$. Our methods will act as post-processing steps to a pre-specified classification model $f$ that estimates $p(y|x)$. Throughout this paper, we will use $\hat{p}(y|x)$ and $f(x)$ interchangeably. We do not assume anything about how the model $f$ is trained. In particular, it need not be aware of the cost function $\mathcal{L}$.

The ultimate goal is to leverage $x$ to inform decisions via a set $S_{f(x)}$ of suggested labels, accounting for both utility and uncertainty. Thus, for every $x$, we would like to produce a set of labels $S_{f(x)}$ that fulfills the following two properties:

1. For a random sample $(x, y)$ drawn from $P$, it must be true that $y \in S_{f(x)}$ with probability at least $1 - \alpha$ [1]. This condition must be fulfilled without any assumptions on the correctness of $f$ or the distribution $P$.

2. $\mathbb{E}_{(x,y) \sim P}[\mathcal{L}(S_{f(x)})]$ is minimized. This quantifies that, on average, the set $S_{f(x)}$ is optimal from an utility-theoretical perspective.

Standard conformal methods guarantee the first property and our goal in this paper is to target the second (cost function minimization) as well while still maintaining such coverage guarantees.

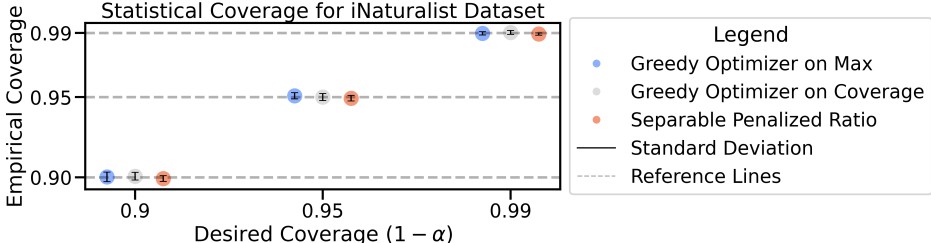

Figure 2: Empirical statistical coverage on the iNaturalist dataset across various loss functions. The observed coverage is close to the expected preset value of $\alpha$ for all 10 trials per loss function. We see a similar behavior for all the other datasets.

---

[1]This is not a conditional statement, instead is a high probability event over the joint distribution of tuples $(Y, X)$

## 3 METHODS

We start by considering a simplified setting in which the cost function has a particular separable structure, which we can leverage to precisely characterize to our problem. We then consider nonseparable loss functions and extend such ideas to the general setting.

### 3.1 SEPARABLE LOSS

As a motivating example, consider a cost function in the medical domain where a healthcare worker must diagnose a condition based on a set of tests. Each test has an associated penalty related to costs, patient discomfort, and procedure complexity. Notably, the penalty of each test does not depend on the presence of other tests in the set. Formally, we say that the cost function $\mathcal{L}$, is separable if it can be decomposed as $\mathcal{L}(S_x) = \sum_{y \in S} \ell(y)$, where $\ell(y)$ is a cost function defined on individual labels rather than on the set of labels as a whole.

One starting point for algorithm development in this setting, is the standard conformal prediction toolbox, in particular, the *split conformal* algorithm. Split conformal uses a held-out calibration set to provide a model-agnostic post-processing step which generates prediction sets with statistical coverage at a predefined level. While this method naturally addresses the first (1) of the two conditions we seek from our prediction sets, it raises an important question: can we leverage *split conformal* to also ensure a low loss $\mathcal{L}$?

At the core of the *split conformal algorithm* is the definition of a *non-conformity score* $s(x, y)$. Larger scores encode worse *agreement* between $x$ and $y$ (Angelopoulos & Bates, 2021). Naturally, the selection of a non-conformity score is key because the prediction sets will be a function of it. There has been recent work (Angelopoulos & Bates, 2021) exploring the choice of a particular score as an engineering question; in general, the non-conformity scores are a function of the estimates $\hat{p}(y|x)$ thus leveraging the notion of similarity between the pair $(x, y)$ captured by the model. The *split conformal algorithm* works by outputting sets $S_x = \{y : s(x, y) \leqslant \hat{\tau}_{1-\alpha}\}$ where $\hat{\tau}_{1-\alpha}$ is the empirical $\left\lceil \frac{(n+1)(1-\alpha)}{n} \right\rceil$ quantile of a independent draw of size $n$ of nonconformity scores, such draw is known in the literature as *the calibration set*.

Returning to our goal of producing prediction sets with low cost function $\mathcal{L}$, one option is to incorporate the loss of the prediction set $\mathcal{L}(S_{f(x)})$ into the score $s(x, y)$. This can be done explicitly if the loss $\mathcal{L}$ is separable as $\mathcal{L}(S_{f(x)})$ can be decomposed into a penalty $\ell(y)$ for each individual $y$. Formally, let $\pi(X) = \{\sigma_1(x), ..., \sigma_k(x)\}$ the permutation that orders the estimates $\hat{p}(y_i|x)$ from greatest to lowest, define $\rho(x, y) = \sum_{i=1}^r \hat{p}(y_{\sigma_i(x)}|x)$ where $y_{\sigma_r(x)} = y$ and define $L(y)$ as $\sum_{i=1}^r \ell(y_{\sigma_i(x)})$. Particularly, $\rho$ is the nonconformity score used in the popular "adaptive prediction sets" algorithm Romano et al. (2020); as labels are added from most to least likely until the true label $y$ is reached, $\rho$ computes a running total of the amount of -estimated-probability mass contained in the set. Notably, $L$ keeps a running total of the cost function incurred when adding elements to the prediction set in that same order. Our proposed nonconformity score is simply $s_\lambda(x, y) := \rho(x, y) + \lambda L(y)$, where $\lambda > 0$ is a hyperparameter. After applying the split conformal algorithm on a held-out calibration set, the resulting prediction sets for a given $\lambda$ are given by $S_{f(x)}^\lambda := \{y : s_\lambda(x, y) \leqslant \hat{\tau}_{1-\alpha}^\lambda\}$ where $\hat{\tau}_{1-\alpha}^\lambda$ is the empirical $\left\lceil \frac{(n+1)(1-\alpha)}{n} \right\rceil$ quantile of an exchangeable draw of size $n$ the scores $s_\lambda(x, y)$ . Note how a higher value of $\lambda$ skews the distribution of the scores; for any pair $(x, y)$ the relevance of the similarity captured by the estimate $\hat{p}(y|x)$ is going to have less weight than the total loss of the *proxy* prediction set $\{y_{\sigma_1(x)}, ..., y_{\sigma_k(x)}\}$ captured by $L(y)$.

The question of how to select the appropriate $\lambda$ remains. One possible approach is to perform a grid search for the optimal value using a separate validation/test split distinct from the calibration dataset. Since maximizing utility is the primary objective, this hyperparameter tuning strategy indeed will converge to the true sought parameter as summarized by the following proposition.

**Proposition 1.** *Let $\lambda \in \mathcal{H}$ where $\mathcal{H}$ is a finite set, assume as well that for every $\lambda$ and every $x$, $\mathcal{L}(S_{f(x)}^\lambda(x)) \leqslant B$. Let $\hat{\lambda}$ be the $\lambda$ that minimizes $\frac{1}{n} \sum_{i=1}^n \mathcal{L}(S_{f(X_i)}^\lambda)$, for an iid draw of size $n$ from $P$. Let $\lambda^*$ be the $\lambda$ that minimizes the population level quantity $\mathbb{E}[\mathcal{L}(S_{f(X)}^{\lambda^*})]$. Fix any $\delta \in (0, 1)$ then with probability at least 1 - $\delta$*

$$\left| \mathbb{E}[\mathcal{L}(S_{f(X)}^{\hat{\lambda}})] - \mathbb{E}[\mathcal{L}(S_{f(X)}^{\lambda^*})] \right| \leqslant 2B\sqrt{\log\left(\frac{2|\mathcal{H}|}{\delta}\right)\frac{1}{2n}}$$

A consequence of the previous preposition is that $\lambda^*$ can be learnt via empirical risk minimization. That being said, doing so makes the estimated $\hat{\lambda}$ dependent on the data. Thereby breaking the exchangeability assumption between a new test point $(X, Y)$ and the original sample. This assumption is crucial for applying the split conformal algorithm to ensure valid statistical coverage. Consequently, after learning $\hat{\lambda}$, the quantile $\hat{\tau}_{1-\alpha}^{\hat{\lambda}}$ must be recalculated using an independent sample.

Taken all together, we then proposed the following algorithm. Firstly, split the data in three folds: a validation set, a test set, and a calibration set. For each value of $\lambda$ in $\mathcal{H}$ estimate the quantile $\hat{\tau}_{1-\alpha}^{\lambda}$ on the val set. Then, we estimate the cost function on the test set and then select the $\lambda$ with the best test loss ($\hat{\lambda}$). Finally, re-estimate the quantile $\hat{\tau}_{1-\alpha}^{\hat{\lambda}}$ on the fresh calibration set to guarantee valid statistical coverage.

Although our proposed sample splitting impacts the learning rate stated in Proposition 1, the final rate of convergence remains $O(\frac{1}{\sqrt{n}})$ relative to the total sample size.

### 3.1.1 HYPERPARAMETER-FREE SOLUTION

The prior solution depends on a hyperparameter $\lambda$ that must be tuned. We also have no formal guarantee that the optimal prediction sets for our problem can be expressed via the penalized family of nonconformity scores. However, it is possible to develop a score derived from a more principled approach that does not require the selection of any hyper-parameters. As a starting point to derive this score, suppose that there is oracle access to the true estimand $p(y \mid x)$. Then, conditions 1 and 2 can be explicitly stated as the following program:

$$\min_{S \subseteq X \times \mathcal{Y}} \mathbb{E}_X[\mathcal{L}(S_X)] \text{ s.t } P(Y \in S_X) \geqslant 1 - \alpha \tag{1}$$

Where $S_x$ are the fibers of $S$ over $x$, formally, $S_x = \{y : (x, y) \in S\}$. This program, under the assumption of a separable loss, has a closed-form solution via the Neyman-Pearson lemma. Define $H = \{(x, y) : \frac{p(y|x)}{\ell(y)} \geqslant t_\alpha\}$ where $t_\alpha$ is the $\alpha$ quantile for $\frac{p(y|x)}{\ell(y)}$. That is, $H(x)$ is the prediction set given by all labels where the ratio of probability to loss exceeds a threshold chosen to ensure coverage.

**Proposition 2.** *The set $H$ minimizes $\mathbb{E}_x[\mathcal{L}(S_X)]$ while fulfilling $P(Y \in H(X)) \geqslant 1 - \alpha$*

We note that Sadinle et al. (2019) characterized solutions for the special case where the loss function is the size of the set (i.e., all labels have equal cost) ; here we provide a general solution for arbitrary separable losses.

Since we do not have access to the ground truth $p(y|x)$, we instead use $f$ as a plug-in approximation for $p(y|x)$ in the set $H$ defined above. Since $f$ may not be correct, we may no longer have the correct coverage level, which we correct with the following conformalization step. Define $s(x, y) = \frac{\hat{p}(y|x)}{\ell(y)}$, let $\hat{\tau}_\alpha$ be the empirical $1 - \alpha$ quantile of an observed iid draw of scores $s(x, y)$. Define $S_{f(x)} := \{y : \frac{\hat{p}(y|x)}{\ell(y)} \geqslant \hat{\tau}_\alpha\}$. We refer to this hyperparameter-free method as the *Separable Penalized Ratio* throughout this work. We can guarantee using standard arguments that the separable penalized ratio algorithm satisfies statistical coverage, as demonstrated in Figure 2 and formalized in the following proposition.

**Proposition 3.** *Let $s(x, y)$ and $S_{f(x)}$ be defined as above. Then*

$$1 - \alpha \leqslant P(Y \in S_{f(X)}) \leqslant 1 - \alpha + \frac{1}{n+1}$$

### 3.2 NON-SEPARABLE LOSSES

Unfortunately, not all relevant set losses $\mathcal{L}$ are separable. In real-world scenarios, we often want to define losses that depend on interactions among the elements in the set. For example, consider our running example of medical diagnosis, where we may prefer prediction sets in which the labels come from a small number of categories within a domain-specific hierarchy, representing more clinically similar diagnoses. This can be formalized as $\mathcal{L}(S_{f(x)}) = \sum_{C \in \mathcal{C}} \mathbb{I}(S_{f(x)} \cap C)$ where $\mathcal{C}$ is a set of (possibly overlapping) categories. In this context, we prefer prediction sets which intersect with as few categories as possible. This loss is not separable because there are diminishing penalties to including a new label in the set once a given category is already covered (as discussed formally in the appendix).

While loss separability was instrumental to the family of non-conformity scores proposed as a solution for the separable case, we now present a conformal-based approach for the non-separable case. Drawing inspiration from the separable scenario, we introduce a non-conformity score that generalizes the one outlined in Proposition 1. The key idea is to linearize the non-separable cost function by summing the marginal increase in loss when adding each additional element to the prediction set.

Let $\pi(x) = (\sigma_1(x), \ldots, \sigma_d(x))$ be the permutation that orders $\hat{p}(y_i|x)$ from greatest to smallest. Define $\rho(x,y) = \sum_{i=1}^{k} \hat{p}(y_{\sigma_i(x)}|x)$, where $y_{\sigma_k(x)} = y$ (this coincides with the homonymous quantity defined in the previous section). Now, let $S_i = \{y_{\sigma_j(x)} : j \in [i]\}$ be the set containing the first $i$ labels in the permutation and define $L(y) = L(S_{\sigma_k(x)}) = \sum_{i=1}^{k}(\mathcal{L}(S_i) - \mathcal{L}(S_{i-1}))$ as the sum of the marginal gains obtained from adding elements using the order given by the permutation $\sigma$. The proposed non-conformity score is then given by $s(x,y) = \rho(x,y) + \lambda L(y)$.

As in the separable case, this heuristic generates sets that ensure coverage while maintaining a low loss. The motivation for selecting $\lambda$ remains consistent with that of the separable case. We can also employ grid search hyperparameter tuning on a validation dataset, as the approach outlined in Proposition 1 does not rely on the separability of the loss.

The definition of the previously presented score, circumvents the non-separability of the loss by fixing the order in which to add labels to the prediction set (based on the confidence score outputted by the model $f$), after which the loss can be decomposed into a series of marginal increments.

### 3.2.1 HYPERPARAMETER-FREE SOLUTION

Just as in the separable loss case, we would prefer an approach that does not rely on hyperparameter tuning and which can benefit from structural information about the optimization problem at hand. The main draw back from the penalized conformal predictor proposed for the non-separable case is that the fixed order (which does not depend on the loss) upon of which it is based may be suboptimal if it does not adapt to the combinatorial structure of the cost function. To address this issue, let us revisit the program in Equation 1 and suppose that we have access to the true estimate $p(y \mid x)$. As in the separable case, it would be ideal to solve this program. However, since the separability assumption no longer holds, it is unclear whether a closed-form solution exists. Suppose instead that we have access to an algorithm which can optimize over the cost function $\mathcal{L}$. Then, we could solve the following optimization problem separately for each value of $x$ to produce the prediction set $S_x$:

$$\min_{S_x \subseteq \mathcal{Y}} \mathcal{L}(S_x) \text{ s.t. } \sum_{y \in S_x} p(y|x) \geqslant 1 - \alpha$$

The prediction set $S_x$ defined by solving this problem for each $x$ would correspond to a feasible solution to the problem in Equation 1. Moreoever, it would minimize the cost function on each instance. However, it actually attempts to offer a stronger coverage guarantee: a conditional coverage guarantee where $S_x$ contains the true label for every $x$, instead of the marginal guarantee in condition 2. Unfortunately, guaranteeing conditional coverage is in general impossible without access to the ground truth $p(y|x)$ (Foygel Barber et al., 2021; Vovk, 2012), so we will be unable to implement this solution exactly.

Our proposal is to use the optimization oracle to solve the above optimization problem at each test instance with the *plugin* estimates $\hat{p}(y|x)$ and then conformalize the resulting sets to provide at least marginal coverage guarantees. While there are many ways to implement this meta-algorithm (depending on the optimization approach which is appropriate to any given cost function $\mathcal{L}$), we focus here on a particular implementation via an efficient greedy algorithm which we find works well for many losses in practice.

Suppose that $\mathcal{L}(S) \leqslant M$. Consider the order in which labels are chosen by the following greedy algorithm for (Leskovec et al., 2007) solving the plug-in optimization problem:

$$S_{f(x)}^{i+1} = S_{f(x)}^{i} \cup \left\{ \arg\max_{y \in \mathcal{Y}/S_{f(x)}^i : \hat{p}(y|x) \leqslant \alpha - p(S_{f(x)}^i)} \frac{M - \mathcal{L}(S^i \cup \{y\})}{1 - \hat{p}(y|x)} \right\} \tag{2}$$

where $p(S_i) := \sum_{y \in S_i} \hat{p}(y|x)$. We then define a non-conformity score as $s(x, y) = \sum_{i=1}^{k} \hat{p}(y_{\sigma(i)} \mid x)$, where $\sigma$ is the permutation that orders the labels according to the greedy construction of the set $S_{f(x)}^{k}$ outlined in equation 2 where $k$ is such that $y = y_{\sigma(k)}$. Our final algorithm will simply run split conformal on this nonconformity score. The crucial difference from the previous approach is the order in which elements are added to the prediction set, which is calculated by simulating the above greedy algorithm on each instance (instead of simply sorting the predicted probabilities). This procedure has standard marginal coverage guarantees:

**Proposition 4.** *Let $\tau_{\alpha,n}$ be the empirical $\left\lceil \frac{(n+1)(1-\alpha)}{n} \right\rceil$ quantile from an iid draw of size $n$ of the scores $s(x, y)$. Define $S_{f(x)} = \{y : s(x, y) \leqslant \tau_{\alpha,n}\}$. Then for any $(x, y) \in P$,*

$$1 - \alpha \leqslant P(y \in S_{f(x)}) \leqslant 1 - \alpha + \frac{1}{n+1}$$

## 4 EXPERIMENTS

### 4.1 SETUP

We conduct a series of experiments to evaluate our proposed methods for both separable and non-separable loss cases, with the following setup.

**Datasets and models.** We evaluate our algorithms for both separable and non-separable settings using the following four datasets: CIFAR-100 (Krizhevsky & Hinton, 2009), iNaturalist (Van Horn et al., 2018), ImageNet (Deng et al., 2009), and the Fitzpatrick dataset (Groh et al., 2021). The iNaturalist dataset consists of a large collection of images of living organisms, with labels organized according to a subtree of the natural taxonomic hierarchy. Our classifier is trained to predict the lowest level of abstraction in this hierarchy, which is the *class* of the organism (here, 'class' refers to the taxonomic rank). These classes are further grouped into *phyla*, which in turn belong to broader taxonomic categories known as *kingdoms*. The Fitzpatrick dataset contains 16,577 clinical images, annotated with both skin condition and skin type labels based on the Fitzpatrick scoring system, along with two additional aggregated levels of skin condition classification derived from the skin lesion taxonomy developed by Esteva et al. (2017). Training a classifier on the Fitzpatrick dataset is particularly challenging (Groh et al., 2021), with good models having top-1 accuracy hovering around 30%. In order to study how the performance of our methods varies with the quality of the underlying classifier, we also include a cleaned version of the dataset where we excluded classes with a validation accuracy below 0.30, retaining 64% of the dataset (reducing the number of classes from 114 to 49) and increasing model accuracy to 0.5. In all experiments, we use EfficientNet-B0 (Tan, 2019), with $\alpha$ fixed at 0.1, as our underlying classifier to estimate the conditional probability $p(y|x)$. We report the median of the means for the loss function of interest across different validation and calibration data samples for each dataset. Additionally, we present the statistical coverage conditioned on set size(Angelopoulos et al., 2020a), in order to investigate that the marginal coverage guarantee does not mask unwanted behavior, such as providing perfect coverage for a specific subgroup of the population at the expense of others. Finally, we report the average value of $\hat{p}(y|x)$ for the true label $y$ as a function of the set size $S_{f(x)}$, which is a proxy for adaptability of our methods (Angelopoulos & Bates, 2021).

**Cost functions.** In the separable case, the total loss was defined as the sum of penalties assigned to the constituent elements of a given set. We applied a uniform random assignment from the set $\frac{i}{4} : i \in [4]$ to each label across all datasets. In the non-separable case, we evaluate two different loss functions over the label spaces: maximum distance and coverage loss. Each of these losses is defined in terms of a pre-specified hierarchy. For the iNaturalist and Fitzpatrick datasets, we use the expert-defined hierarchies that accompany each dataset. For CIFAR-100 and ImageNet, we synthetically generate the hierarchy using hierarchical clustering on the representations obtained from the classifier's final layer. The maximum distance is defined as the largest pairwise graph distance along the hierarchy between any two elements within a predicted set. The coverage loss is defined as the number of categories that are intersected by a predicted set, where each category corresponds to a group of labels at the second-to-last level of the hierarchy. Both losses capture notions of homogeneity within the predicted sets with respect to the hierarchies.

**Comparison against other forms of uncertainty quantification** Our methods, being conformal-based, do not rely on a precise estimation of $p(y|x)$. However, to benchmark against other approaches to uncertainty quantification (many of which rely on properly estimating $p(y|x)$) we conducted experiments to compare our method against predictions sets obtained from Bayesian estimates of $p(y|x)$. We chose Bayesian due to its

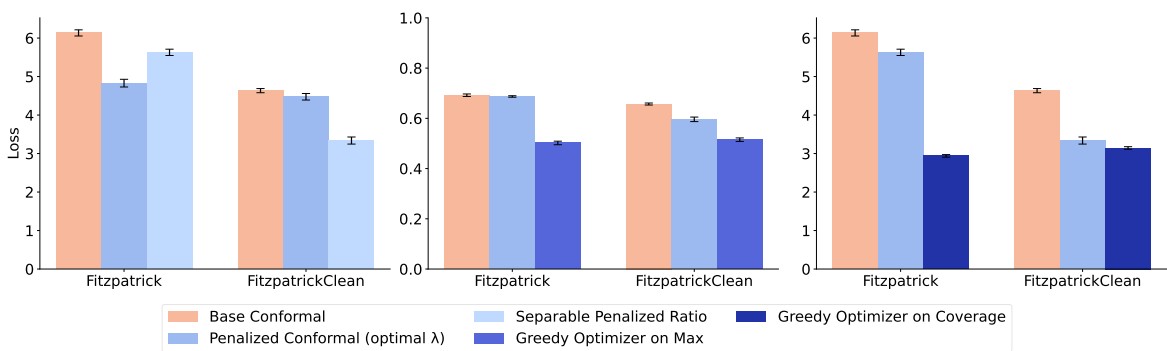

Figure 3: Comparisons between different methods for the 1) separable loss case using the sum of penalties in each set as the loss metric (left), 2) non-separable loss case using maximum penalty in the set as the metric (middle), and 3) hierarchy coverage function loss case using the intersection of the set with each branch of the hierarchy as the metric (right). Numbers are the median-of-means across 10 different runs. See Appendix D for all datasets.

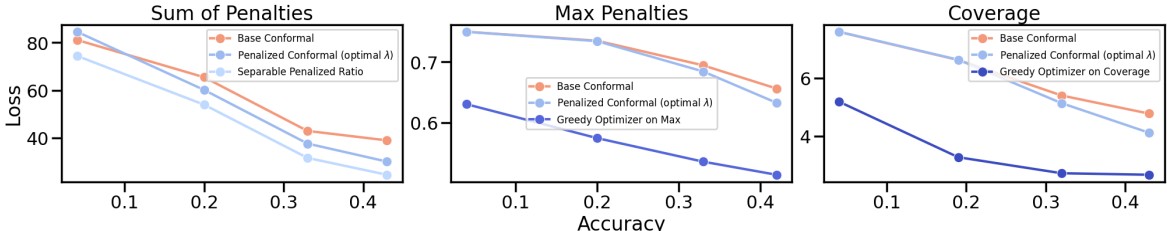

Figure 4: Measured impact of base model accuracy on downstream optimization values on all losses considered for the cleaned Fitzpatrick dataset. Shows median-of-means across 10 different runs for each configuration. Results demonstrate that all methods improve with a higher quality model—i.e., a model with higher accuracy.

widespread adoption within the uncertainty quantification community. The results can be found on the appendix in Figure 6.

## 4.2 RESULTS

In Figure 3 (and 7 on the appendix) together with Tables 4, 5, 6, we observe that our methods have significantly lower cost function than the base conformal method in all considered datasets. This validates our hypothesis that engineering a non conformity score that aligns with a downstream loss results in an utility aligned conformal predictor. In the separable case, we observe that our method outperforms the baseline conformal prediction sets, both when using the Separable Penalized Ratio and Penalized Conformal algorithms (Table 4 and Figure 3). For the non-separable case, we also observe that our method outperforms base conformal prediction for both considered loss functions (Table 5 and Figure 3). Additionally, in Table 1 (and Tables 14, 13 in the Appendix), we demonstrate how our methods successfully provide valid coverage across all set sizes, ensuring robust coverage that is relatively uniform across all predictions. Similarly, Figure 5 (as well as in Figure 8 in the Appendix) illustrates how our methods achieve adaptability, yielding larger sets for instances where the model is less confident in its predictions. These results demonstrate that our framework indeed provides a principled approach to take advantage of latent hierarchical abstractions, enhancing an agent's ability to navigate complex problems, as illustrated in Figure 1.

We further note that, penalized conformal methods, when appropriately tuned, tend to outperform the other methods we tested on CIFAR100, ImageNet, and iNaturalist, likely due to the large dataset sizes. This can be attributed to the fact that both the separable penalized ratio and the greedy optimizer approach assume the Bayes classifier is known, whereas penalized conformal methods do not. However, we posit that for the FitzPatrick dataset, there was not enough data to reliably estimate the optimal $\lambda$, making our principled methods perform better and a suitable off-the-shelf solution for an initial iteration on this dataset.

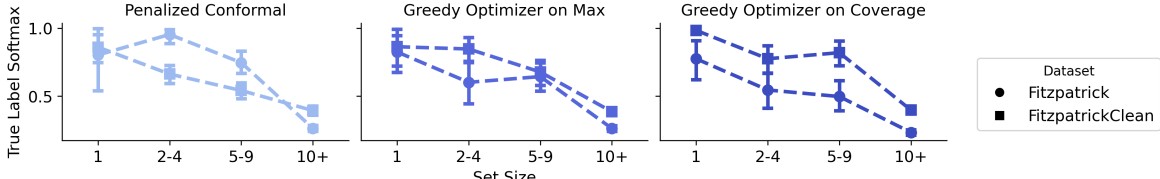

Figure 5: Mean and standard deviation from the predicted softmax score for the true class as a function of predicted set size. The downward trend indicates that the predicted set size increases with the difficulty of the classification task for the model, highlighting the relationship between model uncertainty and set size.

| Size | Base Conformal | | Penalized Conformal | | Greedy Coverage | | Greedy Max | |
|------|------|------|------|------|------|------|------|------|
| | count | coverage | count | coverage | count | coverage | count | coverage |
| 1 | 46 | 0.87 | 55 | 0.87 | 29 | 1.0 | 22 | 0.82 |
| 2 to 4 | 77 | 0.95 | 196 | 0.91 | 50 | 0.96 | 37 | 0.89 |
| 5 to 9 | 102 | 0.88 | 219 | 0.89 | 48 | 0.98 | 36 | 0.92 |
| 10 to 49 | 588 | 0.95 | 383 | 0.90 | 677 | 0.98 | 713 | 0.96 |

Table 1: Conditional statistical coverage by set size for the cleaned version of the Fitzpatrick dataset and a target coverage of $1 - \alpha = 0.9$. Penalized conformal method results are reported for the optimal $\lambda$ value. We also included the number of samples that belong in each size bucket. The average set sizes for all methods and datasets are available in Table 10.

We additionally investigate whether our method can be used in situations where the underlying classifier is noisy. Such robustness can be useful for application settings where practitioners only have access to imperfect, noisy models due to practical concerns (e.g., small amounts of labeled, domain-specific data, etc). In Figure 4, we conduct ablation studies on the Fitzpatrick dataset with a classifier whose training was stopped at different levels of performance to understand the impact of the base model accuracy on downstream optimization values. As expected, we observe that the expected loss of our method decreases with improved classifiers (i.e., having a better base classifier is better). But more importantly, we observe that our method benefits from decision-focused conformalization *even when* the underlying classifier is very noisy, demonstrating that our method outperforms base conformal methods at any level of accuracy of the underlying classifier. This demonstrates that even with noisy classifiers, our methods will still produce prediction sets with higher utility and desired coverage.

## 5    CONCLUSION

We introduce methods that extend traditional conformal prediction to take into consideration utility functions that represent consequences of real-world actions and outcomes. The approach can be harnessed to provide decision-makers with guidance in the form of high-confidence prediction sets that contain homogeneous elements with respect to an external utility metric, ensuring that the actions taken based on these sets are effective or, at the very least, not significantly different in their outcomes according with the utility metric.

We illustrate the operation of our methods via a real-world case study in dermatology diagnosis. In this context, the utility metric reflects homogeneity concerning the hierarchy of dermatological pathologies. That is, good sets are those that ideally form subsets within this established hierarchy. Our method produces prediction sets with valid statistical coverage and a coherent clinical interpretation. The sets generated correspond to subsets of these hierarchical abstractions, aligning with, and potentially supporting, diagnostic and therapeutic reasoning. Our method outperforms existing approaches, and appear robust to challenges such as scarce and noisy data, as demonstrated by the performance on the Fitzpatrick dataset. We hope that our work will provide a valuable stepping stone for future research that leverages uncertainty quantification to support decision makers in high-stakes settings.

**Reproducibility Statement** We are committed to making our research reproducible, allowing future work to build upon our findings. All code and instructions necessary to reproduce all experiments and results is available

at [hidden]. Further, all artifacts (i.e., data, model scores, metadata) and hyperparameter configurations required for running the experiments is also publicly released at [hidden].

## 6 AKNOWLEDGMENTS

We thank Naveen Raman for comments on the manuscript. This work was supported in part by the AI2050 program at Schmidt Sciences (Grant #G-2264474) and the AI Research Institutes Program funded by the National Science Foundation under AI Institute for Societal Decision Making (AI-SDM), Award No. 2229881. Carlos Patiño's presentation of this paper at ICLR 2025 was financially supported by the Amsterdam ELLIS Unit.

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

## A  APPENDIX

### A.1  PROOF PROPOSITION ONE

*Proof.* Formally, Let $P$ be absolutely continuous to $\nu := \mu \times \mu'$, where $\mu$ is the Lebesgue measure and $\mu'$ is the counting measure. Let $H = \{(x, y) : y \in H(x)\}$ and $H(x) = \{y : y \in H(x)\}$.

$$
\begin{aligned}
\mathbb{E}[\mathcal{L}(H(x))] &= \int_{\mathcal{X}} \mathcal{L}(H(x))p(x)dx \\
&= \int_{\mathcal{X}} \sum_{y \in H(x)} \ell(y)p(x)dx \\
&= \int_{H} \ell(y)p(x)dx \\
&= \int_{H} \ell(y)p(x)dx \\
&= \int_{H} \ell(y)d\nu
\end{aligned}
$$

Therefore, as a consequence of the Neyman-Pearson Lemma, we can guarantee that the set $H$ that minimizes $\int_H \ell(y)d\nu$ while fulfilling $\int_H dP \geqslant 1 - \alpha$ is

$$H = \{(y,x) : \frac{p(y|x)}{\ell(y)} \geqslant t_\alpha\}$$

Where $t_\alpha$ is the $\alpha$ quantile for $\frac{p(y|x)}{\ell(y)}$. ∎

### A.2 PROOF PROPOSITION TWO

*Proof.* It is worth notice that for both the maximum and coverage cases $\mathcal{L}(S_\lambda(x)) < B$ for every $\lambda$ and every $x$ (different constant in each case though). Then, If the set of parameters $\mathcal{H}$ is finite, then combining Hoeffding's and the union bound:

$$P(\sup_\lambda |\hat{\mathcal{L}}(\lambda) - \mathbb{E}[\mathcal{L}(\lambda)]|) \leqslant 2|\mathcal{H}|exp(-2n\epsilon^2/B^2)$$

Thus, if $e_n = \sqrt{\frac{B^2 log(2|\mathcal{H}|/\delta)}{2n}}$, then $P(sup_\lambda |\hat{\mathcal{L}}(\lambda) - \mathbb{E}[\mathcal{L}(\lambda)]| \geqslant \epsilon_n) \leqslant \delta$ and thus with probability at least $1 - \delta$:

$$\mathbb{E}[\mathcal{L}(\hat{\lambda})] \leqslant \hat{\mathcal{L}}(\hat{\lambda}) + \epsilon_n \leqslant \hat{\mathcal{L}}(\lambda^*) + \epsilon_n \leqslant \mathbb{E}[\mathcal{L}(\hat{\lambda^*})] + 2\epsilon_n$$

∎

### A.3 PROOF PROPOSITION THREE

*Proof.* It follows as a consequence of the independence of the calibration data points and the application of the split conformal algorithm Angelopoulos & Bates (2021); Vovk et al. (2005) ∎

### A.4 PROOF PROPOSITION FOUR

*Proof.* It follows as a consequence of the independence of the calibration data points and the application of the split conformal algorithm Angelopoulos & Bates (2021); Vovk et al. (2005) ∎

### A.5 PROOF COVERAGE FUNCTION IS NOT SEPARABLE

*Proof.* Suppose is separable, i.e, $\mathcal{L}(S) = \sum_{y \ inS} \ell(y)$ for some $\ell$. In particular this would imply that $\mathcal{L}(\{y\}) = \ell(y)$. Let $C$ be one of the categories used to define $\mathcal{L}$, then:

$$\mathcal{L}(C) = \sum_{C_i \in \mathcal{C}} \mathbb{I}(C \cap C_i)$$
$$= 1$$

However, $\mathcal{L}(\{y\}) = \ell(y)$, thus $\mathcal{L}(C) = |C|$, a contradiction if $C$ has more than one element.

∎

## B EXPERIMENTAL DETAILS

The training details for the neural network classifier by dataset are:

| Dataset | Epochs | Batch Size | Learning Rate | Validation Fraction | Vision Backbone |
|---------|--------|-----------|---------------|---------------------|-----------------|
| Fitzpatrick | 120 | 256 | 0.001 | 0.1 | EfficientNet_B0 |
| Fitzpatrick Clean | 120 | 256 | 0.001 | 0.1 | EfficientNet_B0 |
| iNaturalist | 15 | 256 | 0.001 | 0.1 | EfficientNet_B0 |
| CIFAR100 | 32 | 256 | 0.001 | 0.1 | EfficientNet_B0 |
| ImageNet | 1 | 256 | 0.001 | 0.1 | EfficientNet_B0 |

Table 2: Hyperparameters used to train the classifiers for each dataset. The configurations are identical except for the number of training epochs.

| Dataset | # of Classes | Dataset Size | Base Model Accuracy |
|---------|--------------|--------------|---------------------|
| Fitzpatrick | 114 | 8289 | 0.37 |
| Fitzpatrick Clean | 49 | 4709 | 0.54 |
| CIFAR100 | 100 | 50000 | 0.71 |
| iNaturalist | 51 | 100000 | 0.90 |
| ImageNet | 1000 | 50000 | 0.72 |

Table 3: Details about the datasets we use to calibrate and evaluate the sets. For the iNaturalist dataset, we use the labels up to the class level. We divide each dataset into calibration, validation, and test sets with a split of 50-25-25% of the dataset respectively.

For both separable and non-separable conformal-based methods, the grid used for tuning $\lambda$ was $\{0.001, 0.01, 0.1, 1, 10\}$.

## C  BAYESIAN COMPARISON

Using variational inference, we trained an EffcientNet Tan (2019) where the last linear layer of the net had a Gaussian prior $N(0, 1)$ on its weights. The differentiable objective used to train this *Bayesian neural network*, was the negative evidence lower bound and at each training step the weights and the posterior distribution (being more precise the variational approximation) were updated accordingly. Finally for the prediction phase, for each instance $x$ we draw 100 samples from the posterior distribution to produce estimates of $\hat{p}(y|x, \theta)$, subsequently we then produce a Monte Carlo approximation of $\hat{p}(y|x)$. The neural network was trained for 25 epochs using a learning rate of 0.001 and a batch size of 64 on the FitzPatrick dataset Groh et al. (2021).

We now proceed to define the sets $S_{f(x)}$. Let $\pi(x) = (\sigma_1(x), \ldots, \sigma_d(x))$ denote the permutation that orders $\hat{p}(y_i \mid x)$ from greatest to smallest. The prediction set $S_x$ is defined as $S_x = \{y_{\sigma_1(x)}, \ldots, y_{\sigma_k(x)}\}$, where $k$ is the smallest integer such that $\sum_{i=1}^{k} \hat{p}(y_{\sigma_i(x)} \mid x) \geqslant 1 - \alpha$. However, this procedure does not inherently provide statistical coverage. To address this limitation, we conformalized the scores $\sum_{i=1}^{k} \hat{p}(y_{\sigma_i(x)} \mid x)$ by applying our penalized conformal prediction method with $\lambda = 0$ and using the Bayesian estimates $\hat{p}(y|x)$. The results of this procedure are summarized in Figure 6.

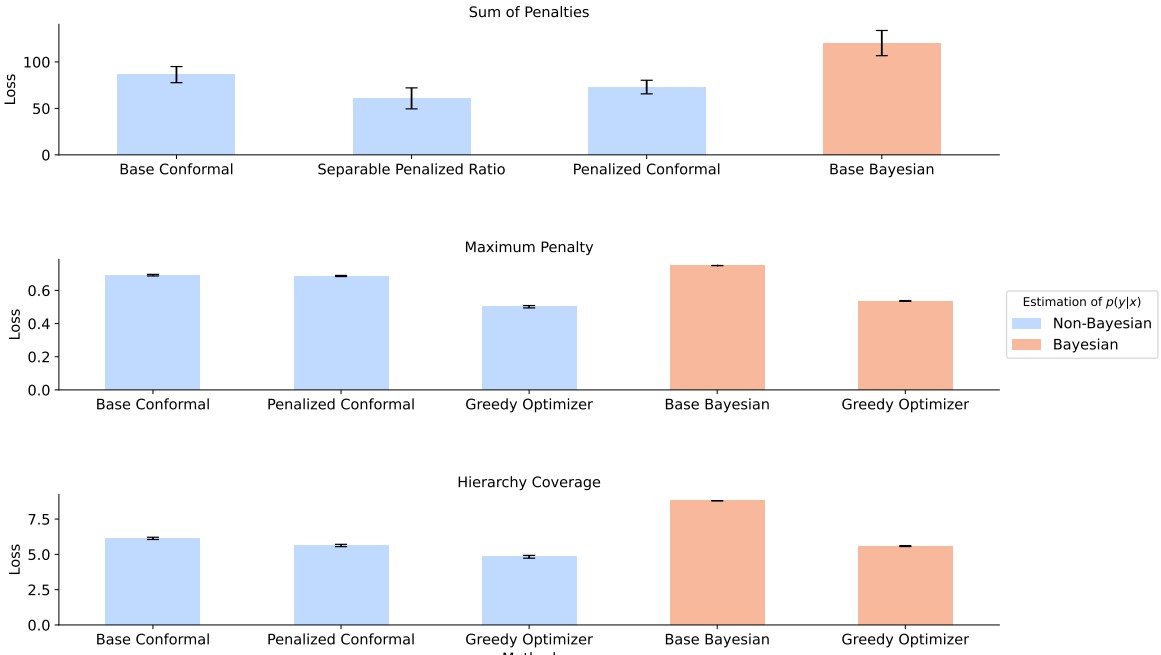

Figure 6: Comparison of our methods when we use a Bayesian model as the source of $\hat{p}(y|x)$. The plots show our methods work well for Bayesian and non-Bayesian approaches because they outperform the base methods in both cases.

| Dataset | Base Conformal | Penalized Conformal (optimal $\lambda$) | Separable Penalized Ratio |
|---|---|---|---|
| CIFAR100 | 29.146 (2.339) | 6.915 (0.477) | 4.442 (0.374) |
| ImageNet | 150.162 (12.889) | 25.553 (2.551) | 5.277 (0.549) |
| iNaturalist | 32.949 (4.982) | 3.855 (0.582) | 2.043 (0.474) |
| Fitzpatrick | 86.373 (8.680) | 72.986 (7.315) | 60.805 (11.300) |
| FitzpatrickClean | 34.356 (3.790) | 22.257 (2.589) | 17.481 (2.236) |

Table 4: Comparison of our proposals for the separable loss case against base conformal predictions for different datasets using the sum of the penalties in each set as the metric. The numbers are the median-of-means across 10 different runs.

**Results** It can be observed that the Bayesian approach not only fails to provide true statistical coverage but also produces prediction sets with a higher average loss. Furthermore, our conformal-based techniques demonstrate remarkable robustness and flexibility by effectively correcting these deficiencies when applied on top of the biased Bayesian estimates of $p(y \mid x)$. This highlights the advantages of our method, as it addresses both the lack of coverage and suboptimal set quality, showcasing its effectiveness once again.

## D EXPERIMENTAL RESULTS

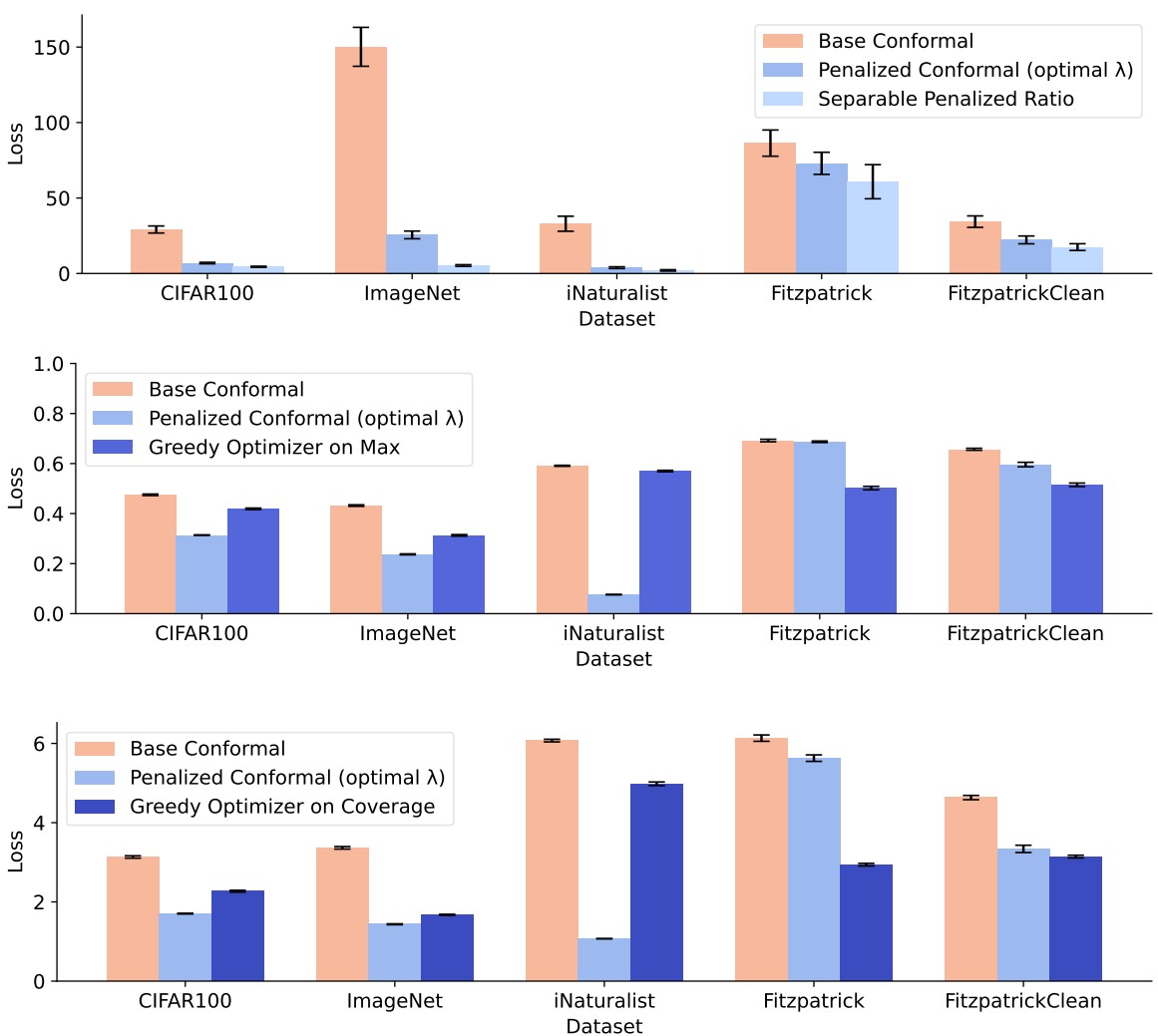

Figure 7: Comparisons between different methods for the 1) separable loss case using the sum of penalties in each set as the loss metric (top), 2) non-separable loss case using maximum penalty in the set as the metric (middle), and 3) hierarchy coverage function loss case using the intersection of the set with each branch of the hierarchy as the metric (bottom). Numbers are the median-of-means across 10 different runs.

| Dataset | Base Conformal | Penalized Conformal (optimal $\lambda$) | Greedy Optimizer |
|---|---|---|---|
| CIFAR100 | 0.475 (0.003) | 0.229 (0.003) | 0.419 (0.003) |
| ImageNet | 0.432 (0.003) | 0.150 (0.002) | 0.313 (0.003) |
| iNaturalist | 0.591 (0.002) | 0.076 (0.001) | 0.570 (0.003) |
| Fitzpatrick | 0.692 (0.005) | 0.678 (0.005) | 0.502 (0.007) |
| Fitzpatrick Clean | 0.657 (0.004) | 0.597 (0.004) | 0.515 (0.007) |

Table 5: Comparison between different methods for the non-separable loss case across different datasets using the maximum penalty in the set as the metric. The numbers are the median-of-means across 10 different runs.

| Dataset | Base Conformal | Penalized Conformal (optimal $\lambda$) | Greedy Optimizer |
|---|---|---|---|
| CIFAR100 | 3.134 (0.030) | 1.054 (0.017) | 2.270 (0.023) |
| ImageNet | 3.365 (0.032) | 0.960 (0.002) | 1.675 (0.013) |
| iNaturalist | 6.072 (0.033) | 1.071 (0.003) | 4.981 (0.046) |
| Fitzpatrick | 6.134 (0.080) | 5.562 (0.057) | 2.940 (0.033) |
| Fitzpatrick Clean | 4.634 (0.054) | 3.338 (0.092) | 3.144 (0.034) |

Table 6: Comparison between different methods for the coverage function loss case across different datasets using the intersection of the set with the hierarchy as the metric. The numbers are the median-of-means across 10 different runs.

| Dataset | 0.01 | 0.01 | 0.1 | 1.0 | 10.0 |
|---|---|---|---|---|---|
| CIFAR100 | 15.039 (1.121) | 6.915 (0.477) | 5.644 (0.812) | 5.846 (0.570) | 6.354 (0.476) |
| ImageNet | 25.553 (2.551) | 8.671 (0.477) | 7.093 (0.963) | 8.389 (1.223) | 8.231 (1.035) |
| iNaturalist | 3.855 (0.582) | 2.818 (0.310) | 2.514 (0.317) | 2.092 (0.362) | 2.371 (0.256) |
| Fitzpatrick | 71.937 (9.291) | 72.986 (7.315) | 83.053 (8.408) | 77.088 (8.263) | 75.293 (9.907) |
| Fitzpatrick Clean | 20.517 (3.350) | 22.257 (2.589) | 24.088 (2.636) | 23.515 (2.816) | 23.780 (2.213) |

Table 7: Comparison between different values of the hyperparameter $\lambda$ for the proposed Penalized Conformal method on separable loss case across different datasets. The numbers are the median-of-means across 10 different runs.

| Dataset | 0.001 | 0.01 | 0.1 | 1.0 | 10.0 |
|---|---|---|---|---|---|
| CIFAR100 | 0.471 (0.003) | 0.443 (0.002) | 0.314 (0.001) | 0.261 (0.003) | 0.229 (0.003) |
| ImageNet | 0.426 (0.003) | 0.374 (0.003) | 0.237 (0.002) | 0.160 (0.001) | 0.150 (0.002) |
| iNaturalist | 0.328 (0.003) | 0.173 (0.001) | 0.076 (0.001) | 0.012 (0.000) | 0.000 (0.002) |
| Fitzpatrick | 0.688 (0.004) | 0.678 (0.005) | 0.687 (0.003) | 0.686 (0.003) | 0.685 (0.004) |
| Fitzpatrick Clean | 0.606 (0.012) | 0.597 (0.004) | 0.603 (0.005) | 0.596 (0.009) | 0.600 (0.010) |

Table 8: Comparison between different values of the hyperparameter $\lambda$ for the Penalized Conformal method on the non-separable loss case across different datasets. The numbers are the median-of-means of the maximum penalty per set across 10 different runs.

| Dataset | 0.001 | 0.01 | 0.1 | 1.0 | 10.0 |
|---|---|---|---|---|---|
| CIFAR100 | 3.079 (0.028) | 2.756 (0.026) | 1.704 (0.009) | 1.096 (0.012) | 1.054 (0.017) |
| ImageNet | 3.279 (0.034) | 2.640 (0.031) | 1.435 (0.012) | 1.053 (0.004) | 0.960 (0.002) |
| iNaturalist | 2.401 (0.022) | 1.432 (0.009) | 1.071 (0.003) | 0.940 (0.001) | 0.902 (0.008) |
| Fitzpatrick | 6.017 (0.070) | 5.562 (0.057) | 5.629 (0.082) | 5.621 (0.062) | 5.613 (0.050) |
| Fitzpatrick Clean | 4.446 (0.104) | 3.587 (0.041) | 3.456 (0.046) | 3.338 (0.092) | 3.419 (0.105) |

Table 9: Comparison between different values of the hyperparameter $\lambda$ for the Penalized Conformal method on the non-separable loss case across different datasets. The numbers are the median-of-means of the coverage penalty per set across 10 different runs.

| Dataset | Base Conformal | Penalized Conformal | Greedy Coverage | Greedy Max |
|---|---|---|---|---|
| CIFAR100 | 14.6 | 5.20 | 22.2 | 23.9 |
| Fitzpatrick | 41.8 | 34.7 | 64.2 | 64.8 |
| FitzpatrickClean | 17.0 | 10.6 | 29.4 | 25.3 |
| ImageNet | 74.8 | 15.1 | 152 | 146 |
| iNaturalist | 17.1 | 1.20 | 23.1 | 22.5 |

Table 10: Average set size by method and dataset. The Penalized Conformal method (with an optimized $\lambda$ parameter) always results in smaller sets than the Base Conformal method. The table also shows that the greedy methods have larger set sizes than the other two methods, which indicates these methods optimize the utility function at the expense of larger sets. However, the predictive value of the sets from the greedy methods is strong, as shown by their conditional coverage and utility function values.

| Size | Base Conformal | | Penalized Conformal | | Greedy Coverage | | Greedy Max | |
|---|---|---|---|---|---|---|---|---|
| | count | coverage | count | coverage | count | coverage | count | coverage |
| 1 | 1041 | 0.94 | 3285 | 0.94 | 1059 | 0.90 | 1025 | 0.94 |
| 2 to 4 | 2665 | 0.97 | 4225 | 0.93 | 2168 | 0.97 | 2145 | 0.97 |
| 5 to 9 | 2152 | 0.99 | 2542 | 0.96 | 1224 | 0.99 | 1198 | 0.99 |
| 10 to 49 | 4948 | 0.99 | 1873 | 0.94 | 5051 | 0.99 | 4563 | 0.99 |
| 50 to 99 | 698 | 0.99 | 15 | 1.00 | 1929 | 0.99 | 2517 | 0.98 |

Table 11: Conditional statistical coverage by set size for the CIFAR-100 dataset and a target coverage of $1 - \alpha$ = 0.9. The results for the penalized conformal method are reported for the optimal $\lambda$ value. We also include the number of samples that belong to each size bucket. We also include the number of samples that belong to each size bucket.

| Size | Base Conformal | | Penalized Conformal | | Greedy Coverage | | Greedy Max | |
|---|---|---|---|---|---|---|---|---|
| | count | coverage | count | coverage | count | coverage | count | coverage |
| 1 | 1446 | 0.99 | 20553 | 0.95 | 1089 | 0.99 | 1122 | 0.99 |
| 2 to 4 | 3776 | 1.0 | 2740 | 0.89 | 784 | 1.0 | 736 | 1.0 |
| 5 to 9 | 3600 | 1.0 | 427 | 0.89 | 3458 | 1.0 | 1440 | 1.0 |
| 10 to 49 | 13516 | 1.0 | 74 | 0.96 | 16353 | 1.0 | 18967 | 1.0 |
| 50 to 99 | 206 | 1.0 | 0 | 0.0 | 908 | 1.0 | 226 | 1.0 |

Table 12: Conditional statistical coverage by set size for the iNaturalist dataset and a target coverage of $1 - \alpha$ = 0.9. The results for the penalized conformal method are reported for the optimal $\lambda$ value. We also include the number of samples that belong to each size bucket.

| Size | Base Conformal | | Penalized Conformal | | Greedy Coverage | | Greedy Max | |
|---|---|---|---|---|---|---|---|---|
| | count | coverage | count | coverage | count | coverage | count | coverage |
| 1 | 967 | 0.95 | 3608 | 0.95 | 894 | 0.94 | 882 | 0.95 |
| 2 to 4 | 1854 | 0.98 | 2657 | 0.94 | 1744 | 0.98 | 1703 | 0.99 |
| 5 to 9 | 1452 | 0.98 | 1744 | 0.96 | 1253 | 0.99 | 1286 | 0.98 |
| 10 to 49 | 2952 | 0.99 | 2866 | 0.95 | 2226 | 0.99 | 2169 | 0.99 |
| 50 to 99 | 1186 | 0.99 | 625 | 0.95 | 992 | 0.98 | 1148 | 0.99 |
| 100+ | 1260 | 0.99 | 327 | 0.95 | 990 | 0.99 | 1159 | 0.99 |

Table 13: Conditional statistical coverage by set size for the ImageNet dataset and a target coverage of $1 - \alpha$ = 0.9. The results for the penalized conformal method are reported for the optimal $\lambda$ value. We also include the number of samples that belong to each size bucket.

| Size | Base Conformal | | Penalized Conformal | | Greedy Coverage | | Greedy Max | |
|---|---|---|---|---|---|---|---|---|
| | count | coverage | count | coverage | count | coverage | count | coverage |
| 1 | 19 | 0.95 | 8 | 0.75 | 24 | 0.79 | 22 | 0.86 |
| 2 to 4 | 42 | 0.90 | 33 | 0.97 | 41 | 0.71 | 31 | 0.84 |
| 5 to 9 | 54 | 0.89 | 84 | 0.92 | 57 | 0.82 | 55 | 0.91 |
| 10 to 49 | 538 | 0.89 | 725 | 0.87 | 131 | 0.85 | 227 | 0.90 |
| 50 to 99 | 459 | 0.92 | 295 | 0.94 | 711 | 0.94 | 298 | 0.91 |
| 100+ | 5 | 1.00 | 0 | 0.00 | 103 | 0.97 | 447 | 0.96 |

Table 14: Conditional statistical coverage by set size for the Fitzpatrick dataset and a target coverage of $1 - \alpha$ = 0.9. The results for the penalized conformal method are reported for the optimal $\lambda$ value. We also include the number of samples that belong to each size bucket.

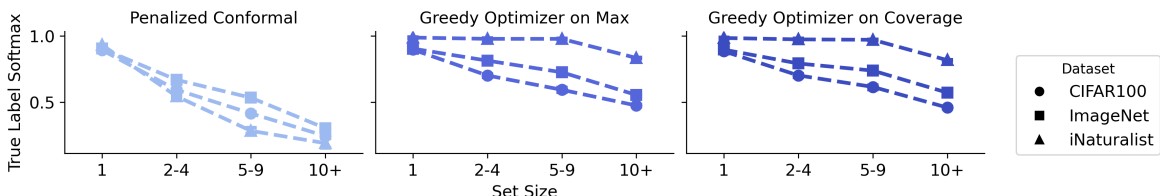

Figure 8: Trend of how the predicted softmax score of the true class decreases as the set size increases. The downward trend indicates the size of the sets grows as the model has more trouble classifying an image.

