# OpenReview forum: "Utility-Directed Conformal Prediction: A Decision-Aware Framework for Actionable Uncertainty Quantification"
_ICLR.cc/2025/Conference — ICLR 2025 Poster_

### Official Review · Reviewer_yMsv · 2024-10-23

**Soundness:** 2
**Presentation:** 3
**Contribution:** 3
**Rating:** 6
**Confidence:** 3

**Summary:**

This paper introduces new methods for uncertainty quantification in classification problems, focusing on conformal prediction techniques that optimize decision loss. Three methods are proposed to address different types of decision losses (separable and inseparable) and are with or without hyperparameters. These approaches retain the coverage guarantee of traditional conformal prediction while achieving significantly lower decision loss, as demonstrated through empirical evaluation on five datasets.

**Strengths:**

- The paper's goal is clear.
- The simplicity of this method, along with the effort to propose a method without introducing a new hyperparameter, will favor its adoption in practical applications.
- Empirical results show notable improvement over the standard conformal prediction method.
- The methods are theoretically well-supported.

**Weaknesses:**

- The discussion of related work is insufficient, focusing exclusively on conformal prediction literature while neglecting other uncertainty quantification methods (quantile regression, Bayesian methods, ensemble methods...). As the goal of the paper is to "Incorporate the decision loss into uncertainty quantification", this omission makes it difficult to assess the significance of the paper beyond conformal prediction.

- Building on the previous point, the paper lacks comparisons with existing methods, even though relevant approaches in conformal prediction appear to address the authors' objectives [1]. Other methods outside of conformal prediction also exist [2], and while they may not offer the same theoretical guarantees, comparing their empirical performance would help assess the impact of the proposed methods. Additionally, traditional conformal prediction could be directly applied to classifiers that focus on minimizing decision loss [3].

- Some sections, such as Figure 2, require improved clarity for better readability.

- The experimental details required to reproduce the results are missing. Specifically, classifiers trained on these datasets can be sensitive to variations in training hyperparameters.

[1] (2022) Conformal Risk Control

[2] (2024) Relaxed Quantile Regression: Prediction Intervals for Asymmetric Noise

[3] (2022) Decision-Focused Learning without Differentiable Optimization: Learning Locally Optimized Decision Losses

**Questions:**

- The authors modify the conformal score by incorporating an additional term to account for decision loss. Wouldn't this make the conformal score less informative and potentially increase the size of the conformal sets? In general, doesn't attempting to minimize decision loss tend to result in larger sets which carry less predictive value? Furthermore, for certain losses, is it possible that adding an extra element to the conformal set could directly reduce the decision loss? To address these concerns, I recommend that the authors report the average size of the conformal sets in their experiments.

- The authors focus on split conformal prediction, but there are other conformal prediction methods available [1,2]. Can the proposed modifications be applied to these other methods, or are they specific to split conformal prediction?

- Following the previous point, the lack of adaptiveness in split conformal prediction is a known issue [1]. How does the proposed method affect conditional coverage? Is the decision loss minimized uniformly across all subgroups, or are there significant discrepancies? Since the authors emphasize medical applications, it seems essential to assess the method's performance on specific subgroups in such contexts.

[1] (2019) Conformalized Quantile Regression

[2] (2022) A Gentle Introduction to Conformal Prediction and Distribution-Free Uncertainty Quantification

---

> ### Author Response · Authors · 2024-11-20
>
> Dear Reviewer  yMsv,
>
> We thank the reviewer for their kind words and enthusiasm for our paper. We address your concerns below.
>
> ### **Expand Related Work**
>
> > The discussion of related work is insufficient, focusing exclusively on conformal prediction literature while neglecting other uncertainty quantification methods
>
> We expanded the discussion in the related work section to include uncertainty quantification (UQ) methods as suggested, and explained why conformal prediction is particularly relevant to our use case. Additionally, we included an extra baseline comparison against variational inference (a Bayesian method) and analyzed its performance relative to our approach.
>
> ### **Additional comparisons with other UQ methods**
>
> > Building on the previous point, the paper lacks comparisons with existing methods… [1] [2] [3]
> > The authors focus on split conformal prediction, but there are other conformal prediction methods available [1,2]. Can the proposed modifications be applied to these other methods, or are they specific to split conformal prediction?
>
> We would like to highlight that while conformal risk control [1] addresses a similar problem, it targets a different setup. CRC allows for controlling the expected value of a set function to a predefined level; if the loss function is the characteristic function of the event that defines coverage (the true label is in the predicted set), then the user will recover true statistical coverage. Thus, CRC allows control over some utility functions but at the expense of coverage guarantees. In contrast, our method is designed to provide both. Moreover, CRC imposes an assumption on the losses it studies—they must monotonically decrease with set size—rendering CRC incompatible with our approach.
>
> The regression setting, where methods like quantile regression might be relevant, presents an interesting extension to our work. However, for now, it lies outside the scope of the topics explored in this draft.
>
> ### **Readability and experimental details**
> > Some sections, such as Figure 2, require improved clarity for better readability.
> > The experimental details required to reproduce the results are missing. Specifically, classifiers trained on these datasets can be sensitive to variations in training hyperparameters.
>
>
>  Thank you for pointing this out! We have added the missing experimental details to Appendix Section B and have improved the readability of Figure 2 to ensure clarity.
> improved the readability of Figure 2 to ensure clarity. We are happy to make further clarifications if this remains a concern!
>
> ### **Adaptability and conditional coverage analysis**
>
> >  the lack of adaptiveness in split conformal prediction is a known issue [1]. How does the proposed method affect conditional coverage? Is the decision loss minimized uniformly across all subgroups, or are there significant discrepancies? Since the authors emphasize medical applications, it seems essential to assess the method's performance on specific subgroups in such contexts.
>
> >  To address these concerns, I recommend that the authors report the average size of the conformal sets in their experiments.
>
> Set-size as a function of entropy of the distribution of labels is the standard to measure how adaptive a conformal based approach is. Hence, we incorporated an analysis of utility by set size in the appendix (and thank the reviewer for the suggestion, as lack of adaptivity is something that can render our method useless). The results demonstrate that our method continues to perform strongly while preserving adaptability. Specifically, our method produces larger sets for challenging examples (instances of $x$ with low-confidence labels), maintaining robustness across diverse scenarios. These results are detailed in Section 4 and appendix D.

---

> > ### Comment · Reviewer_yMsv · 2024-11-25
> >
> > Thank you for your responses and revisions. While some of my concerns have been addressed, my question regarding the method artificially increasing the size of the conformal sets remains:
> >
> > > The authors modify the conformal score by incorporating an additional term to account for decision loss. Wouldn't this make the conformal score less informative and potentially increase the size of the conformal sets? In general, doesn't attempting to minimize decision loss tend to result in larger sets which carry less predictive value? Furthermore, for certain losses, is it possible that adding an extra element to the conformal set could directly reduce the decision loss? To address these concerns, I recommend that the authors report the average size of the conformal sets in their experiments.
> >
> > It would be great if you could address this point.

---

> > > ### Author Response · Authors · 2024-11-25
> > >
> > > ### **Conformal Set Size**
> > >
> > > > I recommend that the authors report the average size of the conformal sets in their experiments.
> > >
> > > We now also report the average size of the conformal sets in Section 4 and Appendix D. The results show that our modified conformal method has a smaller set size than the base conformal prediction method. The greedy methods yield larger sets, but our experiments show that their predictive value is strong, given they respect conditional coverage and outperform other methods in optimizing the utility function.

---

> > > > ### Comment · Reviewer_yMsv · 2024-11-27
> > > >
> > > > Thank you for addressing my question empirically. Based on your response and the results provided regarding counts per set size bin (although I could not locate the figure showing the average conformal set size), I believe it is important to clarify in the revised manuscript that, for the greedy methods, optimizing the utility function comes at the expense of set size.
> > > >
> > > > That said, I recognize that there are scenarios where prioritizing the utility function over set size is justified, highlighting the relevance of this method. I will update my score accordingly.
> > > >
> > > > Could you also specify the exact figure depicting the average conformal set size?

---

> > > > > ### Author Response · Authors · 2024-11-27
> > > > >
> > > > > Dear yMsv,
> > > > >
> > > > > Thank you for your thoughtful comments and review! Our claim was implicit in the results in Table 1, where we show the count for each size bucket. We realized those results did not clearly show the average set size, so we now included **Table 10**, which contains the average set size for all methods and datasets.
> > > > >
> > > > > We hope the updated manuscript addresses your comments. Please let us know if you have additional feedback.

---

### Official Review · Reviewer_arZB · 2024-10-23

**Soundness:** 2
**Presentation:** 2
**Contribution:** 2
**Rating:** 6
**Confidence:** 4

**Summary:**

The paper aims to generate prediction sets that maximize a utility function while having coverage guarantees. It extends standard approaches in conformal predictions with novel conformity scores that integrate the utility of the prediction set. Several utility functions are considered, and the approaches differ depending on whether they are separable or inseparable. The marginal coverage guarantees of conformal prediction can be preserved in both cases. Experiments on three classification datasets and diverse utility functions show that the methods achieve higher utility than the conformal prediction approach that uses the standard non-conformity score.

My evaluation of the paper is mixed. On the one hand, I find the notion of utility of prediction sets very interesting and novel (to my knowledge). On the other hand, I find the paper confusing in some parts, especially in its framing as a "decision-focused" approach, which refers to a stream of literature that provides structured predictions by integrating a constrained optimization problem.

**Strengths:**

The main novelty of the paper is providing prediction sets that maximize a user-specified utility function. If the utility function is the size of the set, we recover the traditional conformal prediction setting. However, the approach is more general. It can consider for instance the hierarchy / categories of the labels in the prediction set. The example in Figure 1 works quite well to illustrate the value of the obtained sets, which are more informative for users.

From a methodological perspective, the algorithms and non-conformity score are quite simple (which is an advantage) and are shown to preserve the coverage guarantees of conformal predictions. The experiments also suggest that they provide good performance.

**Weaknesses:**

I have two main concerns.

First, I find the framing of the paper as a decision-focused approach misleading and confusing. This is for two reasons. It hides the novelty of the paper, which is its focus on prediction sets with utility. It also makes it seems that the paper is focusing on decision-making in the same vein as the cited works of  Mandi et al., 2020; Elmachtoub & Grigas, 2022; Wang et al., 2021, etc. However, this is not the case because (a) the loss function is completely independent of the true label (hence, there is no notion of accuracy / task loss / decision loss / regret of the prediction set) and (b) the prediction sets are unconstrained since the coverage guarantee is achieved by the conformalization procedure. Decision-focused learning typically deals with the challenges of having to output a decision that is both heavily constrained (linear or combinatorial constraints) and provides good task loss thanks to end-to-end training.

An interesting direction to make the problem task-focused / end-to-end would be to consider Equation (1) as the task and train the ML classifier in an end-to-end fashion to minimize this task loss.

I argue that the paper's main focus is conformal predictions with utility and does not have much to do with decision-focused learning. My second main concern is that the current analysis of the prediction sets is a bit light. The experiments do not show how large the prediction sets are (which is a common utility metric in conformal prediction) nor their achieved marginal and conditional coverage. There is likely to be a trade-off between these metrics (size vs. achieved coverage vs. user-specified utility) for having truly informative prediction sets. I would also expect to see benchmarks from the existing literature on conformal predictions, such as the adaptive approaches cited in the paper.

I also have a few minor comments:
- line 53 "*the work to date on decision-focused learning has largely neglected uncertainty quantification, concentrating instead on constructing models to optimize point predictions for specific decision tasks*". This is incorrect. Point forecasts are the focus when the task loss is linear in the predicted parameters. It is not always the case, see among others:\
Qi et al. (2021). Integrated conditional estimation-optimization.\
Chenreddy et al. (2022). Data-driven conditional robust optimization.\
Kallu & Mao (2023). Stochastic optimization forests.\
Sadana et al. (2024). A survey of contextual optimization methods for decision-making under uncertainty.\
as well as the following papers, who all use conformal predictions:\
Chenreddy et al. (2024). End-to-end conditional robust optimization.\
Patel et al. (2024). Conformal contextual robust optimization.\
Yeh et al. (2024). End-to-End Conformal Calibration for Optimization Under Uncertainty.
- line 171. The first property states a conditional coverage guarantee (“*for an instance*”), whereas the objective is marginal coverage (what conformal predictions guarantee and what I believe is shown in Figure 2). It could help to formally introduce the problem using Equation (1), which includes a marginal coverage guarantee.
- The short proofs of Proposition 3 and 4 should be moved to the paper body to clearly show that the coverage guarantee is achieved by the split procedure.

[EDIT:] the authors' response and changes to the manuscript have addressed my main concerns. Therefore, I have raised my score from 3 to 6.

**Questions:**

See weaknesses.

---

> ### Author Response · Authors · 2024-11-20
>
> Dear Reviewer arZB,
>
> We thank the reviewer for their kind words and enthusiasm for our paper. We address your concerns below.
> ### **Framing of the paper**
>
> >I find the framing of the paper as a decision-focused approach misleading and confusing. This is for two reasons. It hides the novelty of the paper, which is its focus on prediction sets with utility. It also makes it seems that the paper is focusing on decision-making
>
> Following the suggestions made by the reviewer we have revised the Introduction section.  Now, it should be more clear that  from decision-focused learning we only draw inspiration as an example of how there are methods that inform decision problems (optimization problems) via machine learning predictions. It is now explicit that our method is not and end to end procedure to train a model both for accuracy and to maximize some utility function, but rather a conformal-based post processing that provides sets with higher utility and marginal coverage. In order to avoid extra confusion we changed the title of the manuscript as well.
>
> ### **Lack of adaptability and conditional coverage analysis**
>
>  > My second main concern is that the current analysis of the prediction sets is a bit light. The experiments do not show how large the prediction sets are (which is a common utility metric in conformal prediction) nor their achieved marginal and conditional coverage.
>
> We have improved our analysis by reporting two extra set of results in the updated version of the manuscript (see Section 4). The marginal coverage guarantees that we provide for our methods could be masking insidious behavior by over covering certain sets and leaving some other group undercover. This is why we now report statistical coverage by set size, thus allowing us to provide a proxy for conditional coverage across instances of different difficulty levels. Additionally, we report the average value of the softmax function for the true label as a function of set size. We use this metric as a measure of the adaptability derived from our conformalized heuristics; ideally, bigger sets should be associated with pairs $(y,x)$ that are hard for the model, i.e, for pairs $(x,y)$ where the value $p(y|x)$ is expected to be low (and thus lead to inaccurate prediction).  The results obtained for both of these new metrics show that our method is robust and adaptive. Our method produces near optimal coverage across all set sizes while also showcasing the desired behavior of yielding larger prediction sets associated with hard examples.

---

> > ### Comment · Reviewer_arZB · 2024-11-21
> >
> > I thank the authors for their response. The authors' answer as well as the changes in the manuscript (reframing, new experiments, new baselines) have adequately addressed my concerns. I raise my score to 6 since I do not see any major limitation remaining.
> >
> > I do not give a higher score as I think that there could be a more in-depth evaluation of the approach. In particular, there could be a discussion and comparison with multi-label classification in which the ML model predicts a set directly instead of the current approach, which trains the ML model to predict a single class and then uses a conformal procedure to get a set. A multi-label approach would allow training the ML model end-to-end on the true utility of the predicted set. However, it would likely lose the coverage guarantee of the conformal procedure.
> >
> > Minor comments:
> > - The authors have not addressed my previous comments regarding missing references (especially the end-to-end conformal ones), ambiguous marginal/conditional sentence on line 177, and moving short proofs to the main body.
> > - Minor LaTeX inconsistencies: \cite vs. \citep line 391-394, also some Tables and Figures are not properly referred to in text (line 450 should be "in Table 1 (and Tables 11, 12 in the Appendix)"; line 765 Figure ?? -> Figure 6.
> > - I still find that using the words "decision" and "decision loss" is misleading. The authors emphasize that the value of the conformal prediction sets is *to inform* the prediction of the current model. There is no decision: only a prediction and a prediction set, which jointly guide practitioners towards their decision. I think calling the approach "structured conformal predictions" or "conformal prediction sets with utility" would more clearly convey this point.

---

> ### Author Response · Authors · 2024-11-25
>
> Dear arZB,
>
> Thank you for the feedback! We address your concerns below:
>
> ### **More in-depth evaluation of the approach (benchmark against an end-to-end procedure to generate prediction sets)**
>
> > I do not give a higher score as I think that there could be a more in-depth evaluation of the approach. In particular, there could be a discussion and comparison with multi-label classification in which the ML model predicts a set directly instead of the current approach, which trains the ML model to predict a single class and then uses a conformal procedure to get a set. A multi-label approach would allow training the ML model end-to-end on the true utility of the predicted set. However, it would likely lose the coverage guarantee of the conformal procedure.
>
> We thank the reviewer for their valuable suggestion. We agree that a comparison with a true decision-focused approach would enrich the discussion in the paper. **To address this, we have added such a comparison in Appendix E**. Unfortunately, the datasets within the paper focus on the single-labeled setting, which did not allow us to implement the *exact* approach suggested by the reviewer. Instead, we have tried our best to develop an approach that would be analogous (i.e., an end-to-end solution). We have updated our draft with this experiment and its results. In our experiments, **under the constraint of achieving the same level of empirical coverage**, our methods outperform the proposed end-to-end solution.
>
> ### **Minor Comments**
>
> Thank you for the feedback and catching these errors! We have made the following updates:
> - Added all missing references (see Section 1.1).
> - Clarified ambiguous sentence on line 177
> - Fixed minor LaTex inconsistencies
> - Updated our title and method name throughout the draft to address the comments on “decision” and “decision loss”
>
> Unfortunately, we did not have enough space to move the short proofs to the main body yet. We plan to do so during camera ready.
>
> We hope this addresses your concerns! Thanks again and please let us know if you have any other feedback.

---

> > ### Comment · Reviewer_arZB · 2024-11-27
> >
> > I thank the authors for their response and changes to the manuscript.
> >
> > In Appendix E, how did you propagate the utility loss through the sorting and thresholding operations? Finding the permutation and selecting the $k$ largest values such that the cumulative probability is greater than $(1-\alpha)$ is not differentiable. More precisely, the gradient of the indicator of the set $S_{f(x)}$ w.r.t. $f(x)$ is zero almost everywhere. This is one of the key challenges of the decision-focused learning literature that is referenced in the paper. Differentiating through this operation likely needs to adapt approaches from the existing literature such as:
> > - Xie et al. Differentiable top-k with optimal transport, NeurIPS 2020.
> > - Blondel et al.. Fast differentiable sorting and ranking, ICML 2020.
> > - Sander et al. Fast, differentiable and sparse top-k: a convex analysis perspective, ICML 2023.
> >
> > Two other limitations of the approach include (i) considering a single dataset and utility metric and (ii) not using conformalized prediction sets. Regarding (ii), what prevents us from training the model to minimize the dis-utility *and* apply the conformal post-processing to produce the sets?
> >
> > I commend the authors for their effort in providing a first approach at end-to-end prediction sets with utility, but I think this analysis deserves more time and broader considerations.

---

> > > ### Author Response · Authors · 2024-11-29
> > >
> > > We use the order of the softmax score to build the prediction sets. The set is such that it has the minimum number of elements that provide at least $1 - \alpha$ “coverage” based on the softmax estimates of the neural network.
> > >
> > > The reviewer is correct regarding the limitations of this additional approach (Just to clarify for readers: this refers to the end-to-end solution and not the main method in our paper). We thank the reviewer for suggesting exploring this additional approach, and we agree this deserves more comprehensive study in its own right, which we leave for future study.
> > >
> > > We once again thank you for your time spent reviewing our work.

---

> > > > ### Comment · Reviewer_arZB · 2024-11-30
> > > >
> > > > Thank you for the response. I understand how the sets are computed. My question is: how did you compute their gradient? Computing the gradient of the loss requires computing the gradient of the sets, which is not trivial.
> > > >
> > > > Since the analysis in Appendix E is limited and I am unsure of its correctness, I suggest removing it from the paper and leaving it for future work.

---

> ### Author Response · Authors · 2024-12-02
>
> We thank the reviewer for their thoughtful feedback and engagement. For the camera-ready version, we will remove Section E of the appendix. Regarding the computation of gradients, we used backpropagation on a loss function that combined the cross-entropy loss with the cost function of interest. Specifically, the cost function was calculated by predicting a prediction set at each step for every element.

---

### Official Review · Reviewer_c9L1 · 2024-11-02

**Soundness:** 4
**Presentation:** 3
**Contribution:** 4
**Rating:** 8
**Confidence:** 5

**Summary:**

This paper tackles a problem in AI decision-making and how to make AI predictions more useful for real-world decisions while maintaining reliability. Instead of just giving a set of possible predictions (like in conformal prediction methods), this work creates prediction sets that make practical sense for decision-makers. The paper suggests a solution that balances probability and cost using tunable parameters and another elegant solution that doesn't need to find this tunable parameter and automatically finds the best balance of likelihood and cost associated with the class.

**Strengths:**

- This is a strong and mathematically sound work that extends traditional conformal prediction work by generating sets of utility functions that capture the associated likelihoods and costs, which is crucial for decision-making.

**Weaknesses:**

- The results section is limited regarding insights provided and evaluated only on 4 common datasets. It would also be a nice insight to see the prediction sets for the experiments as a qualitative analysis.
- It will be interesting to see how this approach works for more challenging datasets with complex hierarchical structures.
- Limited ablation studies
-There is no discussion on the loss functions choices and design beyond maximum distance and coverage

**Questions:**

- The greedy approach overall seems to be doing worse than the approach that uses a learnable $\lambda$ parameter. Could the authors please clarify the advantages of the greedy approach?
- Can you provide theoretical bounds for the performance between greedy optimizer vs optimal solution for non-seperable losses ?
- Is there a relationship between base model calibration and decision loss?
- Did you consult medical experts about the clinical relevance of your prediction sets?

---

> ### Author Response · Authors · 2024-11-20
>
> Dear Reviewer c9L1,
>
> We thank the reviewer for their kind words and their enthusiasm for the paper! We address your concerns below.
>
> ### **Answer to questions**
>
> > Could the authors please clarify the advantages of the greedy approach?
>
> The greedy approach does not require any hyper parameter tuning, thereby functioning as a better off-the-shelf solution.
>
> > Can you provide theoretical bounds for the performance between greedy optimizer vs optimal solution for non-seperable losses ?
>
> Yes we can if the loss $\mathcal{L}$ is submodular, in which case our problem can be described as an instance of sub-modular maximization with a knapsack constrain. These type of problems are known to be efficiently and optimally approximated by greedy-based approaches.
>
> > Is there a relationship between base model calibration and decision loss?
>
> There is no relationship that we are aware of between our losses and a calibrated model. However, this is an interesting question to explore. If we had access to the true values of \( p(y|x) \), our conformalization step would not be necessary. Of course, the Bayes classifier is conditionally calibrated, which is a stronger property than simple calibration, but it would still be valuable to investigate the properties of our method under such conditions.
>
> > Did you consult medical experts about the clinical relevance of your prediction sets?
>
>  One of the authors has a medical background and, although the focus on this paper is on the technical development of an algorithmic framework, we validated with this author the coherence of our method's outputs.

---

### Official Review · Reviewer_2y3o · 2024-11-03

**Soundness:** 3
**Presentation:** 3
**Contribution:** 3
**Rating:** 8
**Confidence:** 3

**Summary:**

This paper presents a new framework that merges conformal prediction with decision-focused learning to create prediction sets that optimize decision-making while maintaining statistical coverage at various levels. The method addresses a gap in current techniques, which often ignore how predictions affect real-world decisions, especially in high-stakes areas like healthcare. The authors propose two algorithms: a penalty-based approach for separable losses and an optimization method for non-separable losses without hyperparameters. They provide theoretical and empirical evidence, including a healthcare case study, to show the effectiveness of their approach.

**Strengths:**

- Originality: Novel integration of decision-focused loss into conformal prediction, addressing a key gap in uncertainty quantification for high-stakes applications.
- Quality: Robust theoretical grounding and empirical results, demonstrating significant decision-loss reduction across multiple datasets.
- Clarity: The paper is well-structured, with a logical flow from motivation to problem formulation, methodology, and results. Also explains previous related concepts well
- Significance: Highly relevant for domains like healthcare, enhancing prediction sets to support actionable, utility-aligned decisions.

**Weaknesses:**

- Lacks comparisons with alternative uncertainty methods like Bayesian inference, limiting context on the framework’s unique advantages.
- Tuning for separable loss is computationally intensive; more efficient tuning methods would improve usability.
- Focuses heavily on healthcare; additional applications in other high-stakes areas would demonstrate broader applicability.

**Questions:**

- For the penalty-based approach in separable losses, can you elaborate on the efficiency of grid-search tuning for large datasets? Are there faster alternatives?
- Could you clarify how the fixed order of adding elements to prediction sets affects performance for complex utility structures? Would alternative scoring or ordering strategies improve flexibility for non-separable losses?

---

> ### Author Response · Authors · 2024-11-20
>
> Dear Reviewer 2y3o,
>
> We thank the reviewer for their kind words and enthusiasm for our paper. We address your concerns below.
>
> ### **Comparison with alternative uncertainty methods like Bayesian inference**
>
> >Lacks comparisons with alternative uncertainty methods like Bayesian inference, limiting context on the framework’s unique advantages
>
> Thanks for the feedback! We have added comparisons against variational inference (a Bayesian method) (see appendix C). The details of how we utilize Bayesian estimates of $p(y|x)$ to construct the predicted sets are now included in the Supplementary Material (see Appendix C). Our new results show that variational methods not only fail to provide true statistical coverage,  but further, are unable to generate high-utility sets. Even when we apply our conformalization approach on top of these estimates, the results are at best comparable to those from our existing methods. This demonstrates that the additional layer of uncertainty quantification provided by variational inference is superfluous for our use case of interest.
>
> ### **Hyperparameter tuning is expensive**
>
> > Tuning for separable loss is computationally intensive; more efficient tuning methods would improve usability
>
> We agree with the reviewer that the required tuning for our conformal-based solution is a drawback, as it increases computational complexity. This is why, as part of the original manuscript, we proposed two methods that **do not require** hyperparameter tuning: our greedy optimizer and the separable penalized ratio method. Both alternatives outperform the base conformal and Bayesian baselines while offering computationally efficient solutions.
>
> ### **Additional examples of potential applications**
>
> > Focuses heavily on healthcare; additional applications in other high-stakes areas would demonstrate broader applicability
>
> - We would like to highlight that our paper considers an additional example outside the medical setting by conducting experiments on the iNaturalist dataset (see Section 4 and Appendix D). This dataset consists of a large collection of images of living organisms, with labels organized hierarchically by taxonomy. Our classifier is trained to predict the lowest abstraction level in this hierarchy, which is the class of the organism (here, ’class’ refers to the taxonomic rank). These classes are grouped into phyla and further into broader taxonomic categories known as kingdoms. Here, the goal is to produce prediction sets that not only contain the true species but also animals that are close in the taxonomic tree (and thus genetically similar). Hence, any prediction about a specimen, even if is wrong, should be at least a very similar animal. Our framework can assist biologists in classifying new species or identifying specific organisms, as confidently determining the type of creature being studied can guide appropriate conservation measures.
>
> - Another application where this framework is useful is within autonomous driving systems. In this setting, we would like to provide a self-driving system the set of possible courses of action to take. However, ensuring the correct action is in the set is insufficient to deem the set as *safe*. Rather we would like *all actions* in the predicted set to be deemed safe, as incorrect choices can be prohibitively costly and detrimental, e.g., a collision or an incident involving a pedestrian. Therefore, it is crucial to provide sets that provide sets of choices that on average yield a low value for $\mathcal{L}(S_{f(x)}) = \max_{s \in \mathcal{S}} \min_{a \in S_{f(x)}} L(a,s)$(for every possible state, the best choice is not very costly) where $L(s,a)$  is a loss capturing the cost of any state-action pair.
>
> ### **Answer to clarification questions**
>
> > For the penalty-based approach in separable losses, can you elaborate on the efficiency of grid-search tuning for large datasets? Are there faster alternatives?
>
> In the paper, we proved that empirical risk minimization converges to the true optimal hyperparameter for a finite set relatively quickly. For infinite sets, this remains an open question. In practice, performing a random grid search is likely the most practical approach.
>
> > Could you clarify how the fixed order of adding elements to prediction sets affects performance for complex utility structures? Would alternative scoring or ordering strategies improve flexibility for non-separable losses?
>
> Using only scores based on $\hat{p}(y|x)$ naturally omits any consideration of the loss. In the non-separable case, the value of the loss depends on the definition of the set itself, leading to a combinatorial optimization problem. In this setting, the order in which elements are added to the set affects both the set's composition and the loss, resulting in different outcomes. Our greedy approach approximates the best possible order, and if the losses are submodular, it can be proven that this procedure is near-optimal.

---

> > ### Author Response · Authors · 2024-11-29
> > **Hope to hear back soon**
> >
> > Dear Reviewer 2y3o,
> >
> > In our response above, we have tried to address all your comments and concerns. To summarize, we have:
> >
> > - Added comparisons with alternative uncertainty methods (Bayesian inference).
> > - Addressed questions on additional applications and how we propose a hyperparameter free solution for our conformal based approach that is computationally efficient.
> >
> > Thank you again for taking the time to review our work and we hope to hear back from you soon. Please let us know if you have any additional questions!

---

### Official Review · Reviewer_rCi4 · 2024-11-03

**Soundness:** 3
**Presentation:** 3
**Contribution:** 3
**Rating:** 6
**Confidence:** 3

**Summary:**

This paper presents a novel framework for decision-focused uncertainty quantification, integrating conformal prediction with downstream decision-making considerations. By introducing decision loss into the prediction process, the authors create a conformal method that offers both standard statistical coverage and improved utility for specific applications, such as healthcare diagnostics.

**Strengths:**

1. The setting of this paper is interesting that taking the decision loss in conformal prediction pipeline.
2. This paper effectively bridges conformal prediction and downstream decision making by incorporating user-specified utility functions, providing both theoretical guarantees and practical applicability. The authors comprehensively address both separable and non-separable decision losses.

**Weaknesses:**

1. The motivation for the study and its illustrative example could be more clearly developed. The authors present the example of the Fitzpatrick dataset to explain their approach, but the rationale for selecting a loss function that reflects the hierarchical homogeneity of dermatologic pathologies remains somewhat unclear. It would benefit the reader if the authors further explained why this hierarchical approach yields a more interpretable clinical result. In addition, I suggest moving this example to the introduction or background section to establish the relevance of the study more clearly.

2. The method relies on applications that require a decision loss function; however, the necessity and specific contexts for using a decision loss function should be more thoroughly explained. The authors claim that their method "outperforms existing approaches" in terms of decision loss; however, since the loss function is central to their conformal prediction adaptation, one would expect it to outperform basic conformal approaches that don't consider decision loss. An explanation of why certain applications requires such a loss function would clarify the broader applicability. For example, in Section 3.1, the authors present a medical example where penalties are associated with the cost and complexity of each test. However, the experimental datasets largely use homogeneity-focused loss functions, so it is unclear why this type of loss should apply to datasets such as CIFAR100. The authors might consider providing additional examples of potential applications and elaborating on appropriate loss functions for each.

**Questions:**

See weaknesses.

---

> ### Author Response · Authors · 2024-11-20
>
> Dear rCi4,
>
> Thank you for the feedback! We address your concerns below.
>
> ### **Further explain motivation and illustrative example**
>
> > "The motivation for the study and its illustrative example could be more clearly developed...In addition, I suggest moving this example to the introduction or background section to establish the relevance of the study more clearly."
>
> The reviewer is correct that we could be more explicit in illustrating how this "hierarchical approach yields a more interpretable clinical result". We have updated our draft to include the following explanation/clarification in the main text (see Introduction). In Figure 1, we observe that although the base conformal prediction method yields a prediction set containing the true label, the prediction set also contains labels (leaves) from **different** parent nodes (benign epidermal, malignant epidermal, non-neoplastic inflammatory). This lacks clinical interpretability as benign vs. malignant diseases need to be treated/dealt with differently. On the other hand, our approach leads to prediction sets that contains labels (leaf nodes) **sharing a common** parent node 'malignant epidermal'. This hierarchical homogeneity of the prediction set aids clinicians in making follow-up decisions (e.g., prioritizing follow-up tests, prescribing treatment, etc.). We are happy to make further clarifications if this remains a concern!
>
> ### **Additional examples of potential applications and elaborating on appropriate loss functions for them**
>
> > "The method relies on applications that require a decision loss function; however, the necessity and specific contexts for using a decision loss function should be more thoroughly explained...The authors might consider providing additional examples of potential applications and elaborating on appropriate loss functions for each
>
> - We would like to highlight that our paper considers an additional example outside the medical setting by conducting experiments on the iNaturalist dataset (see Section 4 and Appendix D). This dataset consists of a large collection of images of living organisms, with labels organized according to a subtree of the natural taxonomic hierarchy. Our classifier is trained to predict the lowest level of abstraction in this hierarchy, which is the class of the organism (here, ’class’ refers to the taxonomic rank). These classes are further grouped into phyla which in turn belong to broader taxonomic categories known as kingdom. Here, the idea is to produce prediction sets that not only contain the true species in it but also animals that are close in the taxonomic tree (and thus genetically similar). Hence, any prediction about a specimen, even if is wrong, should be at least a very similar animal. Our framework can assist biologists in classifying new species or identifying specific organisms, as confidently determining the type of creature being studied can guide appropriate conservation measures.
>
> - Another application example where this framework is useful is within autonomous driving systems. In this setting, we would like to provide a self-driving system the set of posible courses of action to take. However, in this case-study ensuring that the correct action is in the set is not a good enough desiderata to deem the set as \textit{safe}. Rather we would like all choices in the predicted set to be deemed \textit{safe} as well given how incorrect choices can be prohibitively expensive and detrimental, e.g,  a collision or an incident involving a pedestrian. Therefore, it is crucial to provide sets that provide robust-optimization type of guarantees, that is sets of choices that on average yield a low value for $\mathcal{L}(S_{f(x)}) = \max_{s \in \mathcal{S}} \min_{a \in S_{f(x)}} L(a,s)$(for every possible state, the best choice is not very costly) where $L(s,a)$  is a loss capturing the cost of any state-action pair.
>
> We thank the reviewer again for their time and consideration! We are more than happy to address any other concerns they might have.

---

> > ### Author Response · Authors · 2024-11-25
> > **Hope to hear back soon**
> >
> > Dear Reviewer rCi4,
> >
> > In our response above, we have tried to address all your comments and concerns. To summarize, we have added:
> > - Comparisons against other UQ methods (Bayesian inference)
> > - Experiments to measure adaptability and conditional coverage
> > - A comparison against a true decision-focused approach (end-to-end)
> > - The report on the average size of the conformal sets.
> >
> > Thank you again for taking the time to review our work, and we hope to hear back from you soon. Please let us know if you have any additional questions!

---

> > > ### Comment · Reviewer_rCi4 · 2024-11-27
> > >
> > > Thank you for your thorough comments. I appreciate the effort invested in addressing my major concerns. With these improvements, I have decided to increase my evaluation score.

---

### Meta-Review · Area_Chair_xUiC · 2024-12-20

**Metareview:**

This paper considers the problem of integrating conformal prediction based uncertainty quantification with down stream decision-making tasks. To address the gap in current state of knowledge (conformal prediction and decision-focused learning), the paper proposes two solutions. First, a penalty-based approach for separable losses. Second, an optimization method for non-separable losses. Both theoretical and empirical analysis is provided to demonstrate the effectiveness of the proposed approach.

All reviewers' were generally positive about this paper but also asked questions including motivation, additional use-cases and how to use the method for them, comparison with additional baselines, results on prediction set sizes, framing of the paper, conditional coverage analysis, and gradients for sets in an end-to-end approach . The author rebuttal addressed most of these concerns and the paper is revised.

Therefore, I recommend accepting the paper and strongly encourage the authors' to incorporate all the discussion in the camera copy to further improve the paper. Specifically, to increase the impact of the paper, please try to highlight the significance and utility of the proposed method in the context of existing tools including some baselines that came up in the author-reviewer discussion.

**Additional Comments On Reviewer Discussion:**

Reviewers asked questions including motivation, additional use-cases and how to use the method for them, comparison with additional baselines, results on prediction set sizes, framing of the paper, conditional coverage analysis, and gradients for sets in an end-to-end approach . The rebuttal addressed most of these concerns and the paper is revised. Authors' also acknowledged some limitations of their approach in the discussion.

All reviewers' lean towards accepting the paper.

---

### Decision · Program_Chairs · 2025-01-22

Accept (Poster)